

**UNCERTAINTY ESTIMATION OF REGIONALISED DEPTH-DURATION-FREQUENCY CURVES IN**
**GERMANY**
*Bora Shehu[1], Uwe Haberlandt[1]*
[1] Institute of Hydrology and Water Resources Management, Leibniz University Hannover Germany
*Correspondence to: Bora Shehu (shehu@iww.uni-hannover.de)*
**ABSTRACT:**
The estimation of rainfall depth-duration-frequency (DDF) curves is necessary for the design of several water systems
and protection works. These curves are typically estimated from observed locations, but due to different sources of
uncertainties, the risk may be underestimated. Therefore, it becomes crucial to quantify the uncertainty ranges of such
curves. For this purpose, the propagation of different uncertainty sources in the regionalisation of the DDF curves for
Germany is investigated. Annual extremes are extracted at each location for different durations (from 5mins up to 7days),
and local extreme value analysis is performed according to Koutsoyiannis et al. (1998). Following this analysis, five
parameters are obtained for each station, from which four are interpolated using external drift kriging, while one is kept
constant over the whole region. Finally, quantiles are derived for each location, duration and given return period. Through
a non-parametric bootstrap and geostatistical spatial simulations, the uncertainty is estimated in terms of precision (width
of 95% confidence interval) and accuracy (expected error) for three different components of the regionalisation: i) local
estimation of parameters, ii) variogram estimation and iii) spatial estimation of parameters. First two methods were tested
for their suitability in generating multiple equiprobable spatial simulations: sequential Gaussian simulations (SGS) and
simulated annealing (SA) simulations. Between the two, SGS proved to be more accurate and was chosen for the
uncertainty estimation from spatial simulations. Next, 100 realisations were run at each component of the regionalisation
procedure to investigate their impact on the final regionalisation of parameters and DDFs curves, and later combined
simulations were performed to propagate the uncertainty from the main components to the final DDFs curves. It was
found that spatial estimation is the major uncertainty component in the chosen regionalisation procedure, followed by the
local estimation of rainfall extremes. In particular, the variogram uncertainty had very little effect in the overall estimation
of DDFs curves. We conclude that the best way to estimate the total uncertainty consisted of a combination between local
resampling and spatial simulations, which resulted in more precise estimation at long observation locations, and a decline
in precision at un-observed locations according to the distance and density of the observations in the vicinity. Through
this combination, the total uncertainty was simulated by 10,000 runs in Germany, and indicated, that depending on the
location and duration level, tolerance ranges from ±10-30% for low return periods (lower than 10 years), and from ±15-
60% for high return periods (higher than 10 years) should be expected, with the very short durations (5min) being more
uncertain than long durations.
**KEYWORDS:**
Depth-Duration-Frequency Curves, external drift kriging, local uncertainty, variogram uncertainty, spatial uncertainty,
sequential Gaussian simulation



## 1. Introduction

Design precipitation volumes at different duration and frequencies, also known as Depth-Duration-Frequency (DDF) Curves, are necessary for the design of many water-related systems and facilities. These curves are typically generated by fitting a theoretical distribution to the rainfall extremes (either annual extremes – AMS or extremes above a threshold – POT) derived for specific duration intervals at observed locations. Mostly, a Generalised Extreme Value distribution with three parameters (location, scale and shape) is preferred for such applications (Koutsoyiannis, 2004a, 2004b). An adjustment of the rainfall extremes over different duration intervals is also considered either before fitting the theoretical distribution (as in Koutsoyiannis et al. 1998), or after (as in Fischer and Schumann, 2018). As the fitted theoretical distribution can be used to describe the DDF values only at observed locations, regionalisation techniques are applied to estimate these distributions at unobserved locations. The estimation of a regional distribution based on the index method as proposed by Hosking and Wallis (1997) is one of the most used methods in the literature (Burn, 2014; Forestieri et al., 2018; Perica et al., 2019), followed by the kriging interpolation of the parameters describing these theoretical distributions (Ceresetti et al., 2012; Shehu et al., 2022; Uboldi et al., 2014).

Nevertheless, the procedure for the derivation of DDF curves is subjected to different sources of uncertainty which can affect the confidence level of the estimated design values. Such sources of uncertainties include measurement errors, choice of distribution, short observation length, non-representativeness of point measurements for the spatial dependency of extremes, instationarity due to the climate change etc (Marra et al., 2019b). So far for DDF curves in Germany, there is not objective quantification of the uncertainty, but only approximative guessed tolerance ranges between 10-20% (depending on the return period) that should account for the measurement errors, uncertainties in the extreme value estimation and regionalisation, and for the climate variability (Junghänel et al., 2017). So far in Germany, the tolerance ranges are kept constant throughout duration levels and locations, nevertheless such tolerance ranges are expected to be higher for very short observations and high return periods (Poschlod, 2021) especially for short durations and drier climate (Marra et al., 2017). Therefore, there is a need to perform different simulations in order to quantify the tolerance ranges (uncertainty) dependent on duration, location and return period. In this paper, the focus is on developing a framework that accounts for uncertainties due to short observation lengths and non-representativeness of point measurements. Once a framework is developed, it can be used to investigate the role of distribution choice as in Miniussi and Marra (2021) or the role of future climate as in Poschlod (2021).

In the literature, parametric or non-parametric bootstrapping resampling techniques are used to quantify tolerance ranges of DDF curves. Overeem et al., (2008) was one of the first to include the uncertainty of such curves by including only the uncertainty of GEV parameters estimated by a regional bootstrap procedure (sample variability). In their study, extremes from a homogenous region were pooled together to estimate regional probability distribution, which resulted in a narrower uncertainty range at observed locations. Overeem et al. (2009) proposed a bootstrapping technique where same years for all the observed points were resampled together in order to maintain the spatial dependency of the extremes. Uboldi et al. (2014) went a step further and accounted spatial dependency when performing the bootstrapping for each location: extremes from near observations have a higher probability to be resampled at a specific location than the ones from far away. Typically, the bootstrapping procedures are implemented together with the index-based regionalisation as proposed by Hosking and Wallis (1997). Examples in the literature of such applications, are for instance in Burn (2014) and Requena et al. (2019) in Canada where the uncertainty is computed from the confidence intervals of a parametric bootstrap procedure, or in Chaudhuri and Sharma (2020), Notaro et al. (2015), Tfwala et al. (2017), Van de Vyver (2015) where a Bayesian framework is employed to estimate the uncertainty of DDFs curves at different duration levels. Mostly the uncertainty is derived from bootstrap procedure where the 95% or 90% confidence interval width is used as a measure of precision: as lower the confidence interval width, the more precise are the estimates. However, the spatial structure of



uncertainties is not well considered in the index-based regionalisation: first, no uncertainty of the index itself is considered
and propagated, and second, there is no measure how uncertain the locations further away from observations are.
Therefore, local resampling of extreme values (to account for sample variability) are not enough to describe the spatial
structure of uncertainty, instead spatial simulations are needed. Alternatively, remote sensing data, i.e. satellites or weather
radar data, provide spatially continuous indirect measurements of rainfall intensities or volumes (Marra et al., 2019b).
However, their shortcomings are related to the short available dataset, the inability of the remote sensing dataset to capture
accurately intensities, and lack of a true observed dataset to validate the methods applied. While remote sensing provides
a valuable tool and more research is performed in tackling better the uncertainties, at the moment DDF curves from station
observations represent still the standard procedure, and hence a method to estimate the spatial structure of uncertainties
based on these observations is required.
In kriging, when regionalising from point values, the variance of the estimations can be used as a measure of the
uncertainty for un-observed locations. This estimation can either be parametric (multi-Gaussian process) or non-
parametric (indicator kriging). It is widely accepted that the kriging system can capture only the local uncertainty and not
the spatial one, and moreover it fails to preserve the high spatial variability of the target variable (Cinnirella et al., 2005;
Deutsch and Journel, 1998; Goovaerts, 1999b, 2001; Lin and Chang, 2000). As stated in Liao et al. (2016) the spatial
uncertainty is more important (bigger) than the local uncertainty. Therefore, solutions for the estimation of the spatial
uncertainties in geostatistics are stochastic simulations with equiprobably realisation of the target variable in space. The
main assumption of the stochastic simulations is the generation of equiprobable realisations in space while maintaining
certain global statistics of the target variable; for instance, the histogram of the observed values and the semi-variogram
(herein referred as variogram for simplicity) - which describes the spatial dependency of the variable variance on the
distance between the observations. The stochastic simulations present a trade-off: on one side they provide more spatial
variable fields than kriging (which is known for its smoothening properties), and on the other side, because the goal is to
maintain the global statistics, may suffer from larger errors at the local scale. Examples of different stochastic simulations
are the sequential Gaussian simulations (SGS) (Cinnirella et al., 2005; Emery, 2010; Ersoy and Yünsel, 2009; Gyasi-
Agyei and Pegram, 2014; Jang, 2015; Jang and Huang, 2017; Liao et al., 2016; Poggio et al., 2010; Ribeiro and Pereira,
2018; Szatmári and Pásztor, 2019; Varouchakis, 2021; Yang et al., 2018), sequential indicator simulations (SIS) (Bastante
et al., 2008; Goovaerts, 1999a, 2001; Luca et al., 2007), simulated annealing (SA) (Goovaerts, 2000; Hofmann et al.,
2010; Lin and Chang, 2000), turning bands (TB) (Namysłowska-Wilczyńska, 2015) etc. As seen, the most preferred
stochastic simulation in the literature is the SGS due to its simplicity, followed by the SIS and then by SA. Alternatively
a stochastic random mixing (as stated in Bárdossy and Hörning, 2016) with spatial dependency modelled by Copulas
(Haese et al., 2017) or a collocated cokriging simulation (Bourennane et al., 2007) can also be applied. However,
geostatistical simulations remain the preferred choice in the literature for estimating spatial uncertainty, although the main
application is in the geosciences field, with very few applications in rainfall modelling, and to authors knowledge no
application to the regionalisation of extreme design rainfall. Therefore, geostatistics becomes a useful tool to estimate and
analyse the estimation of DDF uncertainties at observed and un-observed locations. The question which of stochastic
simulations is more appropriate for extreme design rainfall naturally raises.
As stated, because of its simplicity the SGS is a very popular method in estimating spatial uncertainty in geostatistics. In
the SGS approach each simulation is considered a realisation of the multivariate Gaussian process, and hence it is strictly
required for the target variable to be multivariate normal. As discussed in Deutsch and Journel (1998), the testing of the
multivariate normality is a difficult task, which depending on the case at hand, can be very time and computational
expensive and hence is not usually tested. Typically, studies in literature include a transformation to normal distribution
in order to ensure that the target variable is at least univariate normal. Another disadvantage of the normalisation needed





for the SGS application, is that the upper and lower tail of the transformed variable will cause an under/over – estimation
of these values, and hence an extrapolation to lower and upper bounds is required. Contrary, to the SGS, the sequential
indicator simulations (SIS) does not need a prior assumption on the multivariate normality of the target variable and is
more suitable for observed values that do not exhibit bivariate normal properties. The SIS is a conditional simulation
based on the indicator kriging theory, which provides the probability that a location has to exceed a certain threshold. The
number of thresholds considered should be more than 5 but lower than 15 as suggested by Luca et al. (2007). For each of
the selected threshold a variogram is fitted to the portion of the data following under this threshold, and it is used for the
sequential simulation. A disadvantage of the SIS is that, if many threshold classes are presented, order relationship
problems will arise on the obtained realisations (Deutsch and Journel, 1998; Journel and Posa, 1990), which are more
emphasized if empty thresholds are included (Luca et al., 2007). Another disadvantage of the SIS is that mainly it has
been used together with simple and ordinary kriging theory (Deutsch and Journel, 1998), and no application of the SIS in
an external drift or universal kriging has been reported (to authors knowledge) in the literature. Alternative to the SGS
and SIS stochastic simulations, the simulated annealing (SA) can be also implemented to alternate and generate
conditional images of a continuous target variable. The main idea in the implementation of the SA, is a numerical
algorithm which perturbs continuously an image until an objective criterion is reached. The optimization function can
include only one criterion (typically the global statistics) or multiple criteria depending on the application at hand. For
instance Goovaerts (2000) included three criteria: the local estimation of the variable, the observed histogram and
variogram. The advantage of the SA is that no prior assumption of the normality is required (as the observed histogram is
reproduced) and that it allows a degree of flexibility for realisations that doesn't exactly match the objective criteria. On
the other hand, the disadvantages of the SA include the prior selection of the objective criteria carefully and, depending
on the application, the high computational time.
In a previous study, Shehu et al. (2022) investigated different methods and datasets in Germany for the local estimation
of the DDFs from station data, and different regionalisation methods for the estimation of the DDFs at ungauged locations.
Their study revealed that kriging interpolation of long observation records (more than 60 years) with a denser network of
short observations as an external drift delivered best cross-validation results for return periods higher than 10 years.
Therefore, apart from the stochastic simulations that account for the spatial uncertainty, more simulations are needed to
tackle other sources of uncertainties for the estimation of DDF curves: such as sample variability, variogram estimation
and the combination with an external drift. For this purpose, the SGS and SA will be implemented and investigated for
their suitability in generating spatial simulations for DDF curves. Once a best method is chosen for this purpose, different
experiments are conducted based on non-parametric bootstrapping techniques to investigate how each of the uncertainty
component is propagated into the final DDF curves, and if some components are more dominant than others. Lastly, based
on the most important components, a framework for estimating the total uncertainty in regionalised DDF curves (both at
observed and un-observed locations) is proposed.
The paper is organized as following: First, in Section 2 the data and methods for the estimation and regionalisation of
DDF curves is explained (Section 2.1 and 2.2), together with the necessary transformation to normality in Section 2.3 and
testing the bi-Gaussian conditions in Section 2.4. Then an introduction to the main uncertainty sources considered here is
given in Section 3, and the main methods to tackle each uncertainty sources are given in Section 3.1 to 3.3. An overview
of the experiments and how the uncertainty is measured in terms of both accuracy and precision is described in Section
3.4. The results are summarised in section 4, where first a comparison of the two spatial simulations techniques is
investigated (Section 4.1), and later uncertainty results of different experiments for un-observations locations and for the
whole German region are shown respectively in Section 4.2 and Section 4.3. Lastly conclusions and the best framework
to tackle uncertainties for DDF curves in Germany are discussed in Section 5.



**2. Study Area and Data Processing**
The investigation is carried out for Germany, as shown in **Figure 1**, together with the two rainfall measuring networks
from the German Weather Service (DWD) used for the uncertainty analysis, grouped in LS (short for long recording
stations)– tipping bucket sensors with 1min temporal resolution, 0.1mm accuracy, 2% uncertainty and observation lengths
from 40 -80 years, and in SS (short for short recording stations) – digital sensors with 1min temporal resolution, 0.01
accuracy, 0.02-0.04 mm uncertainty and observation length from 10-35 years. An overview of the data from these two
networks is given in Shehu et al. (2022). For both networks, the 1min time steps are aggregated to 5min and then Annual
Maximum Series (AMS) are extracted for each station for 12 durations levels from 5min to 7 days. To avoid the
underestimation of the rainfall depth due to fixed accumulation periods of 5, 10 and 15min, corrections factors of 1.14,
1.07 and 1.04 were used for the AMS of these durations according to the regulations in DWA-531 (DWA, 2012). Next,
as described in Shehu et al. (2022) a jump elimination according to sensor changes is performed (DVWK, 1999) in order
to ensure the stationarity of AMS at most stations for different duration levels.

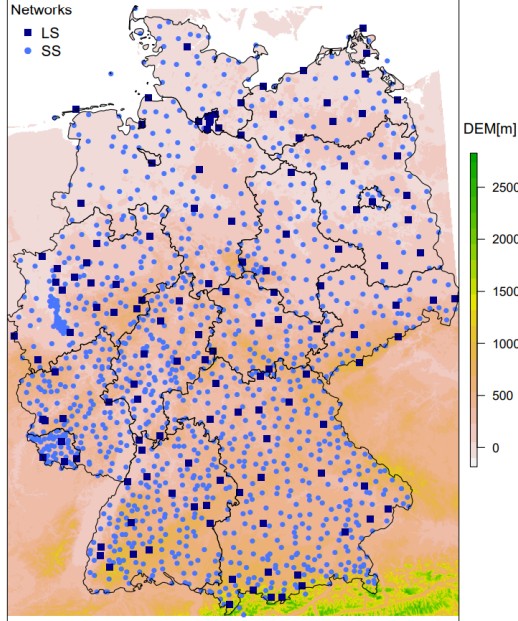

***Figure 1*** *The distribution and location of the two rainfall networks used for the uncertainty analysis of Depth-Duration Frequency Curves in Germany: where LS represents the long and SS the short recording stations. DEM is short for digital elevation model (m) from SRTM.*

***2.1 Extreme Value Analysis***
The local rainfall extreme value statistics describing the DDF curves for each station, are derived in two steps. First, the
intensities of different duration levels are generalised according to the mathematical framework proposed by
Koutsoyiannis et al. (1998) also illustrated in Equation (1):
$$i = i_d \cdot (d + \theta)^\eta,$$ *(1)*
where $i$ is the generalised intensity in mm/h, $i_d$ is the AMS intensity in mm/h at each duration, $d$ is the duration in hours
and $\Theta$, $\eta$ are the Koutsoyiannis parameters optimised for each station. The optimisation of the Koutsoyiannis parameters
is done by minimising the Kruskal-Wallis statistic. Second, a Generalized Extreme Value (GEV) distribution is fitted to





the generalised intensities through the methods of the L-Moments (Asquith, 2021). The GEV is described by three
parameters: location – μ, scale – σ, and shape – γ (with notation according to Coles, 2001) as given in Equation (2). For
a robust estimation of extreme values with return periods of 100 years, the shape parameter was fixed at 0.1. For more
information regarding the choice of generalisation or shape parameter, the reader is directed to our previous study (Shehu
et al., 2022).
$$F(x; \mu, \sigma, \gamma) = exp\left\{ -\left[ 1 + \gamma \frac{(x + \mu)}{\sigma} \right]^{-\frac{1}{\gamma}} \right\}, \qquad \gamma = 0.1 \tag{2}$$

Finally, the local statistics of each station are described by five parameters: three from the GEV distribution (μ, σ, γ) and
two from the intensity generalisation over all durations (θ, μ). Since the shape parameter is fixed at 0.1, only 4 parameters
are regionalised independently from one another using kriging.

### 2.2 Direct Regionalisation (interpolation)

Here a spherical variogram is employed to describe the increment of the variance between any two points of observation
situated at a specific distance h, as per Equation (3). The parameters of the variogram are estimated by of the methods of
the least squares and human supervision.
$$\gamma(h) = c_0 + c \cdot \left( \frac{3h}{2a} - \frac{h^3}{2a^3} \right) \ for \ h \le a \ and \ \gamma(h) = c \ for \ h = a \ , \tag{3}$$

where $c_0$ is the nugget, $c$ the sill and $a$ the range of the variogram. Once the theoretical variogram is known, it can be used
as a basis for regionalising the statistical properties on a 5km$^2$ grid. The regionalisation (or the interpolation) with kriging
is done in two steps, by considering independently the short (SS) and long (LS) recording stations. First, each of the SS
parameters are interpolated with ordinary kriging (herein referred to as OK[SS]) based on the theoretical variogram of
the SS dataset. Second, each parameter derived from the LS dataset is interpolated with external drift kriging KED[LS|SS]
based on the theoretical variogram of LS dataset, whereas the OK[SS] serves as an external drift. The reason for this two-
step procedure, is that the short stations have too little observation years for estimating extremes of high return period,
but still provide useful information about the spatial trends. For more information regarding the choice of this spatial
regionalisation, the reader is directed to our previous study (Shehu et al., 2022).

### 2.3 Data Transformation

A requirement for the spatial simulations (Sequential Gaussian Simulation - SGS), is that the target variable to be
interpolated (in this case each of the 4 parameters), should follow a normal distribution. Following the quantile-quantile
plot, with sample vs normal quantiles, illustrated in **Figure 2**, it is clear that the dataset (both LS and SS) are not normally
distributed, as the extremes (both lower and upper tail) denote clearly from the normal distribution (the dashed continuous
lines). Therefore, in case of a Sequential Gaussian Simulation (SGS) for assessing the spatial uncertainty, a transformation
to normality is required. Deutsch and Journel (1998) propose a normal score transformation based on the empirical
probabilities (Weibull plot position) as indicated in Equation (4).
$$F(x)' = 1 - \left( \frac{k}{n+1} \right) \ and \ x_{norm} = G^{-1}(F(x)'), \tag{4}$$

where $F(x)'$ is the empirical cumulative distributed function calculated based on the descending rank $k$ of input data $x$, $n$
is the number of available $x$-observation, $G^{-1}$ is the inverse function of the gaussian distribution, and $x_{norm}$ is the normalised
input data.





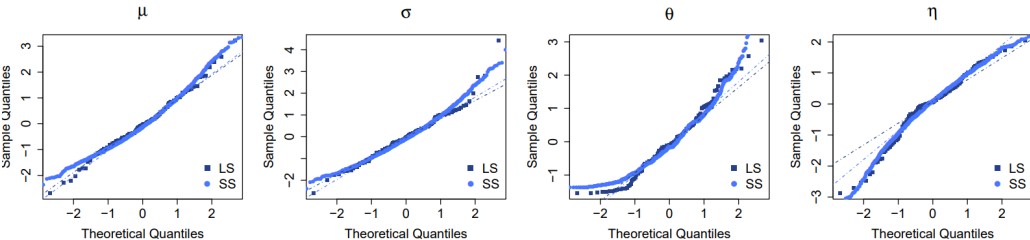


***Figure 2** Sample quantiles of the 4 obtained parameters for both LS and SS datasets in comparison with the theoretical quantiles from the normal distribution. The dashed lines represent the normal quantile lines for a perfect fitting between the sample and the normal quantiles.*

Respectively a back-transformation algorithm is also available to transform back the dataset from the normal to its original
space. However, the back-transformation may be problematic as the tail behaviour will be underestimated by the normal
score and back transformation. An alternative approach to the normal score transformation, is the fitting of the theoretical
cumulative probability functions (CDF) to the original dataset, and perform the transformation from the chosen theoretical
CDF to the normal one. Here, the problem of the choice for tail extrapolation is substituted with the choice of fitting a
theoretical CDF. Through the moment of L-Moments, different theoretical distributions were fitted to the available
datasets, for instance the Wakely distribution (WAK), the Weibull (WEI), the Generalized Normal (GNO) and the
Generalized Extreme Value (GEV) probability distribution. For more information about the CDF and the fitting of the
parameters, the reader is directed to Asquith (2021), Hosking and Wallis (1997). Afterwards the Cramer von Mises
Goodness of Fit test (CSöRgő and Faraway, 1996) is performed to test whether or not the observed data belongs to the
chosen theoretical CDF. The p-value statistics is used to compare the empirical CDF with the theoretical one for each
dataset, in order to select the most adequate theoretical CDF. The results of the p-value statistics from Cramer von Mises
Test are shown in **Table 1**, and they reveal that the parameters of the long stations (LS) are better described by the WAK
distribution, while the parameters of the short stations from the GNO distribution. All the parameters, except the $\theta_{[SS]}$,
exhibit a very large p-value (higher than 0.90). Even though the p-value for $\theta_{[SS]}$ is 0.24, the null hypothesis that the
theoretical distribution describes the current dataset can still not be rejected. To keep a consistent choice between the short
and the long dataset, the GNO was chosen, as the best theoretical distribution for the SS and the second best for LS (shown
in bold letters in **Table 1**).

***Table 1** p-values of Cramer-von-Mises test for testing if the different theoretical distribution fits well to the data. The higher the value, the higher the certainty in accepting the null hypothesis that the chosen CDF describes correctly the data.*

| | Long Station Dataset (LS) | | | | Short Station Dataset (SS) | | | |
|---|---|---|---|---|---|---|---|---|
| CDFs | wak | wei | gno | gev | CDFs | wak | wei | gno | gev |
| $\mu_{[LS]}$ | 0.99 | 0.8 | **0.94** | 0.91 | $\mu_{[SS]}$ | 0.77 | 0.68 | **0.99** | 0.99 |
| $\sigma_{[LS]}$ | 0.96 | 0.8 | **0.9** | 0.85 | $\sigma_{[SS]}$ | 0.85 | 0.39 | **0.980** | 0.95 |
| $\theta_{[LS]}$ | 0.91 | 0.67 | **0.78** | 0.76 | $\theta_{[SS]}$ | 0.24 | 0.15 | **0.24** | 0.2 |
| $\eta_{[LS]}$ | 0.94 | 0.36 | **0.36** | 0.25 | $\eta_{[SS]}$ | 0.52 | 0.83 | **0.91** | 0.27 |

A comparison of these two transformations, normal score according to Deutsch and Journel (1998) and the quantile-
quantile transformation based on fitted theoretical distribution, was performed priory on a cross-validation mode for the
SGS runs in ordinary kriging and external drift kriging. The results of such comparison favoured the quantile-quantile
transformation based on fitted theoretical distributions.



### 2.1 Data Bi-Normality

An additional precondition to run the SGS and assess the spatial uncertainty is the multivariate normality. However as stated in Deutsch and Journel (1998), the data for checking multivariate normality (the tri-variate, quadrivariate and so on) are hardly enough to allow the interference of the corresponding experimental multivariate frequencies. Thus, they suggest that if the bivariate normality conditions are not violated, one can continue with the SGS experiments. Here the bivariate normality is tested by comparing empirical indicator variograms of the normalised parameters sets with the respective ones from a Bi-Gaussian random function that shares the same variogram with the normalised parameter sets. First, a theoretical variogram is fitted to the normalised observed variograms from dataset LS and SS (separately). Next the analytical relation given at Deutsch and Journel (1998) linking the covariance $C_y(h)$ with any normal bivariate CDF value (with mean 0 and standard deviation 1).

$$Prob\{Y(u) \leq y_p, Y(u+h) \leq y_p\} = p^2 + \frac{1}{2\pi} \int_0^{arc \sin C_Y(h)} \exp(-\frac{y_p^2}{1+sin\theta}) \, d\theta, \qquad (5)$$

where $y_p$ in the normal p-quantile of the normal bivariate CDF, and the $C_y(h)$ is the correlogram obtained from normalised LS and SS dataset. For a given threshold $y_p$, the bivariate probability will be:

$$Prob\{Y(u) \leq y_p, Y(u+h) \leq y_p\} = E\{I(u;p) \cdot I(u+h;p)\} = p - \gamma_I(h;p), \qquad (6)$$

with $I(u;p)$ equal to 1 for $Y(u) \leq y_p$ or equal to 0 if otherwise, and $\gamma_I(h;p)$ is the indicator variogram for the p-quantile (corresponding to threshold $y_p$) of the normal bivariate CDF. Three thresholds were chosen for the computation of the indicator variograms that corresponds to 0.25, 0.5 and 0.75 percentiles. Based on Equation (6), the generation of the Bi-Gaussian functions was performed of each set of data independently (short and long) with the GSLIB package. Lastly, the sample indicator variograms for the three thresholds are computed from the observed normalised datasets. The check consists in comparing for each threshold the empirical indicator variogram and the theoretical indication variogram from the normal bivariate CDF.

The obtained indicator variograms are shown in **Figure 3** for empirical data set (in points) and for the Bi-Gaussian functions (in solid lines) of the two datasets (short and long). From **Figure 3** it is visible that the Bi-Gaussian indicator variograms described well the empirical data sets for most of the cases. For instance, the θ and η parameters show a good agreement for the two types of indicator variograms. For the μ and σ parameters the agreement is better for the high thresholds than for the low one (0.25 percentile), where mainly the LS dataset differs more with the Bi-Gaussian indicator variogram than the SS dataset. To a certain degree this is expected, as the LS dataset is much smaller than the SS dataset. Overall, the Bi-Gaussian indicator variograms match well with the empirical ones, and the bivariate normality conditions are not violated. Hence, the SGS can be used for spatial simulation of the parameter sets.

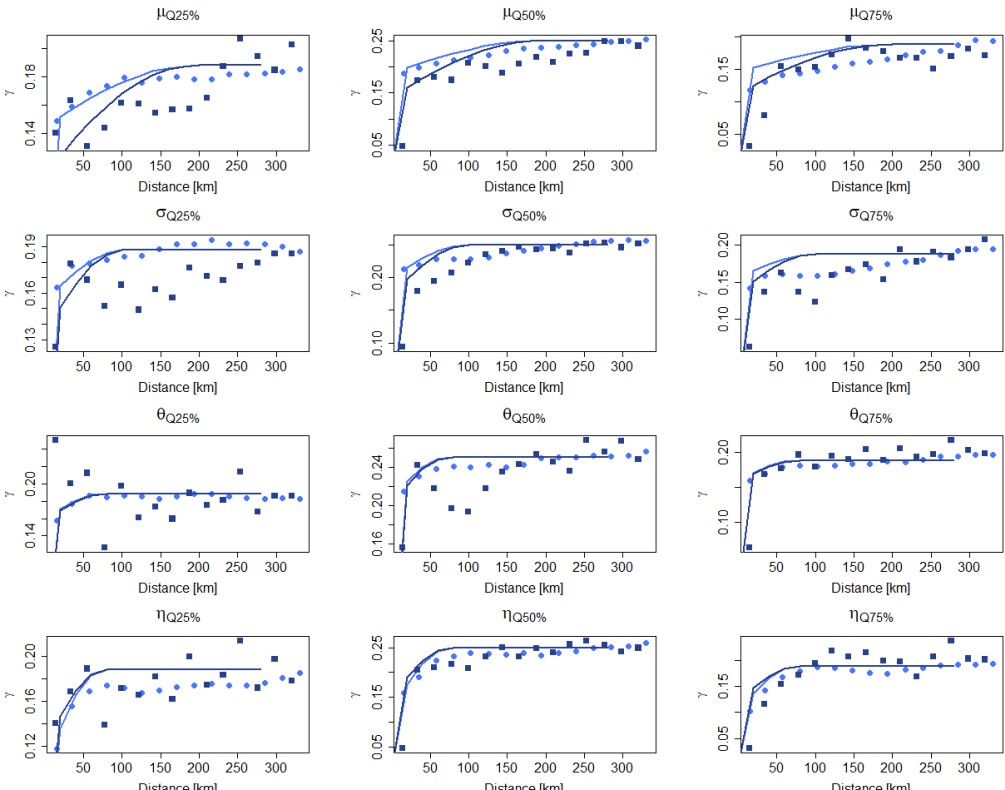

***Figure 3*** *Experimental indicator variograms for the two datasets (SS in light blue, LS in dark blue) for the 4 parameters*
*and their respective fits of the Bi-Gaussian model derived theoretical curves (shown respectively in solid line).*


### 3.   Methods for uncertainty estimation

The regionalisation of the 4 parameters describing the rainfall extreme value statistics, is performed using kriging, as the
best regionalisation method from Shehu et al. (2022). The regionalisation is done primarily with the LS data and using
the interpolation of SS parameters as an external drift. In this procedure, there are several sources of uncertainty that one
should consider for the overall uncertainty, as illustrated in **Figure 4**, which are respectively:
• Sample uncertainty in estimating local extreme value statistics (4 parameters), herein referred to as the local
uncertainty.
• The uncertainty in the external drift which originates from the uncertainty in the estimation of the variogram
based on the SS stations, and from the uncertainty in the regionalisation of the SS statistics. Here, only the latter
is considered, as previous work revealed that this is more relevant than the former.
• The uncertainty in the regionalisation of the LS statistics originating from the estimated variogram from LS
stations, and the uncertainty of the spatial regionalisation (herein referred to as spatial uncertainty).
Overall, the methodologies to tackle these uncertainties can be categorised in three main groups: the local estimation, the
variogram estimation and the spatial simulation (as illustrated in blocks in **Figure 4**). The methodology for uncertainty
estimation on each block is discussed accordingly in the following sections.






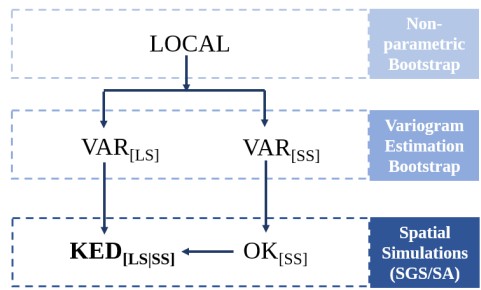

*Figure 4 The main uncertainty sources in the regionalisation of the rainfall statistics for Germany for the selected methodology. Arrows indicate the calculation flow, and the blocks at the right represent the three main methodologies to tackle the uncertainty at each component.*

### 3.1 Local Non-Parametric Bootstrap


A non-parametric bootstrap approach is implemented in order to quantify the sample uncertainty of the local rainfall
extreme value statistics. This means that for each station the AMS are resampled with replacement for the same length of
observations and the local statistics are then derived base on the methodology explained in Section 2.1. This resampling
procedure is run 100 times for each location (either LS or SS), and for each time the parameters describing the local
extreme value statistics are calculated. The resampled parameter-sets are then used as input for the rest of the
regionalisation approach to first investigate the effect of the local uncertainty on the regionalisation output (results shown
in Section 4.2) or their impact on the overall uncertainty of regionalised DDFs curves in Germany (results shown in
Section 4.3).

### 3.2 Variogram Simulations


A non-parametric bootstrap is implemented in the variogram uncertainty, with the precondition that the spatial dependency
between stations is maintained. The whole station dataset (both short and long stations) are grouped together, from which
133 stations are sampled randomly 100 times. For each of the sample, first the empirical variogram is calculated and then
a theoretical spherical one is fitted automatically. Such sampling of variogram, is indirectly accounting the low station
density and the short observation length for the final interpolation of KED[LS|SS]. The obtained variogram simulations
are shown in **Figure 5**. For each of the estimated variogram, the kriging interpolation is performed and in the end its effect
on the final regionalisation output is discussed in Section 4.2.

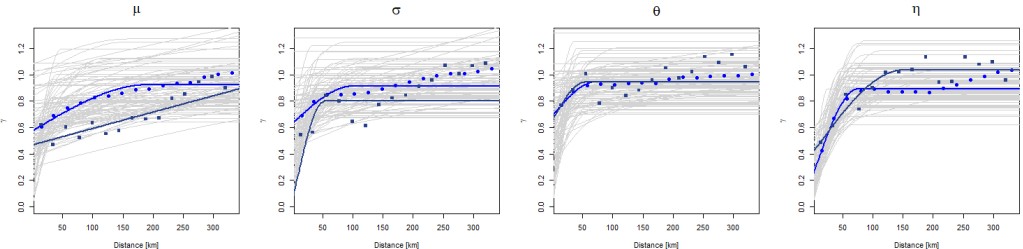


*Figure 5 100 variogram realisations obtained from bootstrapping (shown in grey lines) the station datasets, the empirical variograms as observed by the normalised LS (in dark blue points) and SS database (in light blue points), and the respective fitted theoretical spherical variograms used for the interpolation.*



### 3.3 Spatial Simulations

The uncertainty in the spatial regionalisation is assessed by generating 100 equiprobable realisation of the normalised
parameter sets, where each realisation is honouring the global statistics of the parameter (the spatial mean value and the
variogram). Here a conditional simulation is performed, where these 100 realisations do not only share the global statistics
but as well a set of observed values at certain locations (coinciding with the LS locations).

#### 3.3.1 Sequential Gaussian Simulation (SGS)

The Sequential Gaussian Simulation (SGS) is the most straight forward algorithm for generating such equiprobable
realisation and it is proven to be more robust than other algorithms (Pebesma and Wesseling, 1998). An overview of this
procedure, where a normal continuous variable $z(u)$ is modelled by a Gaussian stationary random function $Z(u)$ is
described as follows (Deutsch and Journel, 1998):
1. A random path is defined that is visiting each node of Germany grid (at 5km$^2$ spatial resolution) once. At each

node $u$, fix the neighbouring conditional locations (either SS for OK[SS] and LS for KED[LS|SS]) and their

observed values $z$, as well as the previously simulated $z$ values at the grid node.

2. Do either ordinary kriging with the normalised short series (OK[SS]) or kriging with external drift with the

normalised long series (KED[LS|SS]) using the respective variograms to estimate the global statistics (mean as

318         per Equation (7) and variance as per Equation (8)) of the Conditional Cumulative Distribution Function (CCDF)

at the random function $Z(u)$ at the location $u$.

$$\mu(u) = \sum_{i=1}^{n} \lambda_i \cdot Z(u_i), \tag{7}$$

$$\sigma^2(u) = C(0) - \sum_{i=1}^{n} \lambda_i \cdot C(u - u_i), \tag{8}$$

where $\lambda_i$ are the weights as estimated by ordinary kriging for OK[SS] and kriging with external drift for

KED[LS|SS], $Z(u_i)$ is the conditional value of the target variable at the $u_i$ location, with $i$ corresponding to

conditional values in the neighbourhood (within a maximum radius of 300km and within the range 12 to 24),

$C(0)$ is the variance and $C(u - u_i)$ the covariance of the normalised dataset.

3. Draw randomly a value from this CCDF as $z'(u)$ and add this simulated value to the conditional dataset.
4. Proceed to the next node, until all nodes are simulated.
The "gstat" package available in R is used to generate such realisation both for the ordinary kriging interpolation of the
SS database (OK[SS]) and for the external drift kriging interpolation of the LS database (KED[LS|SS]) (Pebesma, 2004).
Note that the spatial simulations are always performed on the normal space (normal transformation of the dataset). For
the simulation of the KED[LS|SS] both the input dataset LS and the external drift OK[SS] are as well in the normal space.
A back-transform to the original space is done after each spatial simulation only for the final product KED[LS|SS].

#### 3.3.2 Simulated Annealing Simulations (SA)

Simulated Annealing is an alternative method for generation conditional stochastic images. New images are created by
randomly selected values from the observed histograms, such that global statistics like variogram, marginal distribution,
correlation to a secondary variable are maintained. Unlike the SGS method, no prior assumption of normality is needed,
and hence the observed data (with no prior transformation) can be directly used. An overview of this procedure is found
in (Deutsch and Journel, 1998) and also explained shortly below:
1. An initial image is randomly created by the observed histogram. For nodes where data is observed, the random

values are substituted by the observed ones. Thus, the observed values are exactly reproduced. This image

matches the observed histogram and conditional data, but not the observed variogram.





2.  An objective function is calculated, and a conditional simulation is reached when the objective function is as close as possible to zero. For generation of the external drift spatial information (OK[SS]) only the variogram is used as part of the objective function, while for the final parameter estimation (KED[LS|SS]) additionally the correlation with the external drift is preserved.

$$OF_{OK[SS]} = w_1 \sum_h \frac{[\gamma'(h) - \gamma(h)]^2}{\gamma(h)^2}, and\ OF_{KED[LS|SS]} = w_1 \sum_h \frac{[\gamma'(h) - \gamma(h)]^2}{\gamma(h)^2} + w_2\ [\rho' - \rho]^2 \qquad (9)$$

where $\gamma'(h)$ is the simulated variogram, $\gamma(h)$ the observed variogram, $\rho'$ the simulated correlation and $\rho$ the observed correlation with the external drift, $w_1$ and $w_2$ are weights for the two components (both equal to 5).

3.  If the value of the objective function is not reached, a new image is created by swapping randomly values of pair nodes (not conditioned nodes), and the objective function in recalculated.

4.  If the new objective function is better than the previous one (closer to zero), then the swap is accepted, if not the swap is accepted based on an exponential probability distribution. The parameter of the exponential probability distribution is equal to the temperature in simulated annealing.

$$Prob._{accept} = \begin{cases} 1, & if\ OF_{new} \leq OF_{old} \\ e^{\frac{OF_{old} - OF_{new}}{t}}, & otherwise \end{cases} \qquad (10)$$

where $Prob._{accept}$ is the acceptance probability distribution, $t$ is the temperature (which decreases with each iteration), $OF_{new}$ is the new objective function obtained by swapping a pair of values and $OF_{old}$ is the previous objective function value. As higher the temperature, the higher the probability to selected such unfavourable swaps.

5.  Redo step 3-4, until a maximum number of swaps is reached, or if a maximum number of accepted swaps is reached. If this is the case, the temperature t is reduced by a reduction factor λ.

6.  Redo steps 3, 4, and 5 until convergence is reached or if the maximum number of possible swaps is reached S times. The simulation is then completed and the image is frozen.

The "GSLIB" program from (Deutsch and Journel, 1998) was employed to generate 100 random realisation fields for both the external drift and the interpolation. Note two main differences of the SA with SGS: i) no data transformation and back transformation is required, ii) by fixating a seed number, the random path in SGS is same for all the parameters, while for the SA the random path for each parameter depends on how fast the optimum criteria is reached.

### 3.4 Uncertainty Estimation and Propagation

Based on several simulations, the uncertainty is evaluated only at the locations on the long series (LS) – in total 133 stations. Different experiments are conducted in order to investigate first how the sources of uncertainty are propagating to the final regionalisation of the 4 parameters (experiments 1-4), and how the main sources of uncertainty are interacting with each other to produce the total uncertainty (experiment 5). An overview of these experiments and the sources of uncertainty they consider, is given in **Figure 6** and in **Table 2**. Note that in experiment 5, two uncertainty sources are combined: the local uncertainty from the sampling of rainfall extreme value statistics and the spatial uncertainty from KED[LS|SS] simulations. This means that at experiment 5 for each realisation of the local statistics, both variograms of LS and SS are re-calculated, the OK[SS] is derived and respectively 100 KED[LS|SS] simulations are generated, concluding thus in a total of 10,000 simulations. The bootstrapping of the variograms (VAR[LS|SS]) is left outside of this experiment, because as it is shown in section 4.2, doesn't have a major impact on the regionalisation output. Moreover, as the variograms are re-estimated, different variograms are as well modelled, including the variogram uncertainty indirectly. Here only the combination of local and spatial uncertainty at KED[LS|SS] simulations are included as prior work revealed that this produces the highest uncertainty in terms of precision.





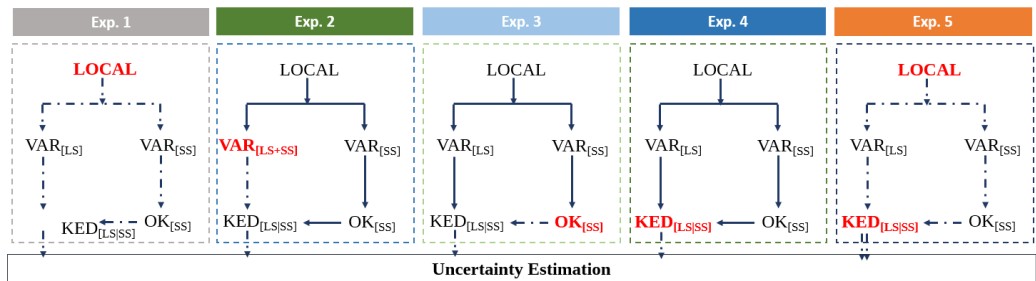


*Figure 6 Different experiments run for the propagation of the uncertainty. The red bold letters indicate the source of uncertainty investigated for each experiment and how it propagates throughout the regionalisation procedure (in dashed arrows). The number of arrows at the experiment 5 indicate different uncertainty sources combined together.*

*Table 2 The description of the uncertainty propagation for each of the experiments shown in Figure 6, and the number of realisations considered for each experiment.*

| Exp. | Explanation | No. of realisations |
|------|-------------|---------------------|
| 1 | For each local re-sampled extreme value statistics, the regionalisation procedure is run. | 100 |
| 2 | For each variogram estimated from LS+SS database, the regionalisation procedure is run. | 100 |
| 3 | For each spatial realisation of the OK[SS], the regionalisation procedure is run. | 100 |
| 4 | For each spatial realisation of the KED[LS\|SS], the regionalisation procedure is run. | 100 |
| 5 | For each local re-sampled extreme value statistics and spatial realisation of KED[LS\|SS] the regionalisation procedure is run. | 10,000 |

For each of these experiments, the final regionalisation step of the 4 parameters (KED[LS|SS]) is run on a cross-validation
mode: which means that each of the LS station is left stepwise outside of the database, and the remaining database is used
to estimate this LS location. The simulations at the LS stations are then used as a basis for the uncertainty estimation of
each parameter separately, and for the final rainfall depth (RD) obtained at specific return periods (T1a, T10a and T100a)
and 12 duration intervals (5, 10, 15, 30, 60, 120, 180, 360, 720, 1440, 2880, 7340 mins). For each LS location, the
uncertainty is estimated based on the experiment simulations using the following criteria:
Normalised 95% Confidence Interval Width:    $nCI95_{width} \; [\%] = 100 \frac{x_{97.5\%} - x_{2.5\%}}{\bar{x}},$    *(11)*
where $x$ represents the simulations of the target variable at a specific location, $x_{97.5\%}$ and $x_{2.5\%}$ are the respective 97.5%
and 2.5% quantile of the $x$ simulations, and $\bar{x}$ is the expected value of $x$ from the simulations of an experiment. The
normalised 95% Confidence Interval Width ($nCI95_{width}$) is a measure of spatial simulations precision: the smaller the
value, the more robust or precise is the estimation method for $x$.
Average Error over all simulations:    $Bias \; [\%] = 100 \frac{\sum_{sim=1}^{nsim} \left( \frac{x_{sim} - x_{obs}}{x_{obs}} \right)}{nsim},$    *(12)*
where $x$ represents the simulation of the target variable at a specific location from the random simulation *sim*, $x_{obs}$ is the
local observed target variable at the specific location, and *nsim* represent the total number of simulations for each
experiment. The average error over all the simulations measures the accuracy of the realisation compared to local input
data. When rainfall depth (RD) is the target variable, one can go one step further and measure how well the realisations
capture the monotonically increase of the RD at different duration intervals for specific return periods, which corresponds





to the evaluation criteria in estimating the best regionalisation method for Germany on our previous study (Shehu et al.,
400   2022).

Percentage RMSE:   $$RMSE_{st,Ta}[\%] = 100 \cdot \frac{\sqrt{\frac{1}{D}\sum_{d=1}^{D}\left(RD_{regio,d} - RD_{local,d}\right)^2}}{\overline{RD_{local}}},$$   (13)
where $Ta$ and $st$ are the respective selected return period and LS location, $RD_{regio}$ corresponds to the regionalised rainfall
depth (with KED[LS|SS]), $RD_{local}$ the locally derived rainfall depth from the normalised GEV function (from Equation
(1) and (2)), the $\overline{RD_{local}}$ is the mean local rainfall depth over all duration levels, and the $d$ is an index indicating the
iteration from $1^{st}$ to D=$12^{th}$ duration interval. Through the Equations (12) and (13) and the cross-validation mode, it is
possible to compare the performance the simulations with the direct regionalisation (i.e. interpolation) from Shehu et al.
(2022), in order to investigate if the simulation methods are appropriate.
**4.   Results and Discussion**
*4.1  Comparison of different models in modelling spatial uncertainty*
Before analysing the propagation of different uncertainty sources, the best method for computing the spatial uncertainty
is investigated. As discussed in Section 3.3 two methods are employed for the generation of 100 equiprobable realisations
both for the drift information (OK) and the interpolation of the long stations with external drift kriging (KED): the
Sequential Gaussian Simulation (SGS) as method 1 and the Simulated Annealing (SA) as method 2. **Figure 7** illustrates
the parameter precision (nCI95$_{width}$ [%]) and accuracy (Bias [%]) of these 100 simulations calculated in cross-validation
mode for each of the long recording locations (in total 133) for both methods. Note that the transformation to normality
is required only for the SGS and not the SA simulations, as the SA simulations are performed based on observed
histograms. The main differences between the two simulation methods are seen in the precision obtained from the 100
realisations (nCI95$_{width}$ – upper row), where the realisations from the SA approach are more precise than the ones from
the SGS approach. The difference in the precision is much higher in the KED[LS|SS] than for the OK[SS] for all the 4
parameters. In terms of parameter accuracy, both methods have similar performance for both OK[SS] and KED[LS|SS],
with SA having slightly higher errors than the SGS and the direct regionalisation (i.e. interpolation) performance
(particularly for the μ and θ parameter). Overall it seems that the SA is more precise than the SGS, nevertheless as the
focus is on Depth-Duration-Frequency curves, the methods should be as well compared in their ability to estimate the
DDF curves. For this purpose, for each cross-validation location, the RMSE [%] was first calculated as per Equation (13)
for each simulation, and then the median over the 100 simulations was obtained. The median RMSE [%] performance for
different return periods for both methods are shown in **Figure 8**. The median RMSE [%] performance obtained by the
SGS method seems to be in accordance with the performance of the direct regionalisation (interpolation) for both OK[SS]
and KED[LS|SS]. In contract, the RMSE [%] performance from the SA simulations are slightly worse than the direct
regionalisation for OK[SS], and much worse for the KED[LS|SS] over all return periods (median up to 5-8% higher).
Even though the SA produces more precise simulations of parameters, it fails to maintain the inter-relationship between
the parameters, causing lower accuracy in the DDF estimation. The SGS on the other hand, keeps the same level of
accuracy like the direct regionalisation (interpolation) but with a lower precision. Since the aim is to keep accuracy as in
the direct regionalisation (interpolation), SGS was chosen as a more suitable method to model the spatial uncertainty.





Also, since the SGS produces a higher range of simulations, the estimated precision, in the end, is more conservative than
the SA procedure.

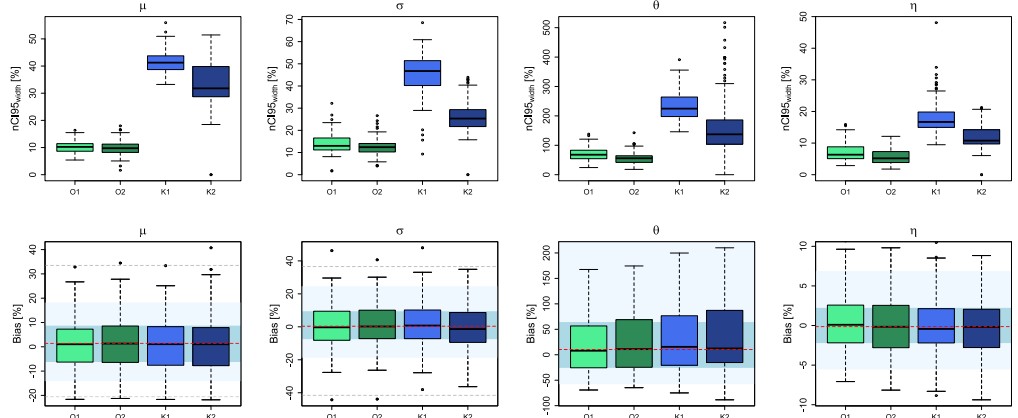

***Figure 7*** *The precision (nCI95$_{width}$ [%]) and accuracy (Bias [%]) of two different spatial simulations methods (1-SGS and 2 -SA) for the drift regionalisation (O) and final regionalisation (K) of the 4 parameters. The boxplots illustrate the performance over the 133 LS locations. The background shades in the lower row illustrate the accuracy of the direct regionalisation (i.e. interpolation) of observed local statistics in a cross-validation mode, where: red dash indicates the median accuracy over all stations, the blue region the inter-quantile range (IQR) of all stations, the light blue region the 95% and 5% quantiles, and the grey dashed lines the maximum and minimum performance over all stations.*


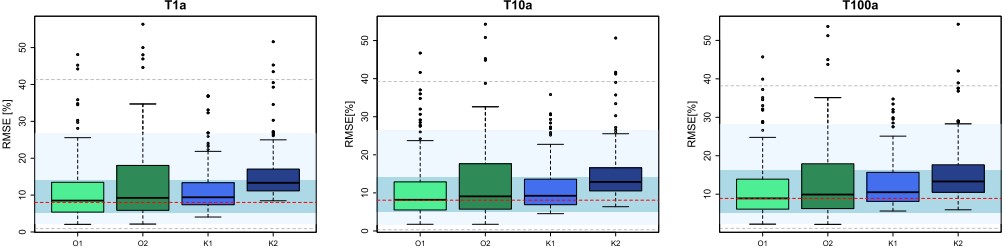

***Figure 8*** *The accuracy (RMSE [%]) of two different spatial simulations methods (1- SGS and 2 - SA) for the drift regionalisation (O) and the final regionalisation (K) of the Depth-Duration-Frequency curves. The boxplots illustrate the median RMSE over the 133 LS locations. The background shades illustrate the accuracy of the direct regionalisation (i.e. interpolation) of observed local statistics in a cross-validation mode, where: red dash indicates the median accuracy over all stations, the blue region the inter-quantile range (IQR) of all stations, the light blue region the 95% and 5% quantiles, and the grey dashed lines the maximum and minimum performance over all stations.*





### 4.2 Effect of different uncertainty components for the estimation of the DDF Curves at ungauged locations

Experiments 1 to 4 were conducted in order to investigate the uncertainty propagation from each component of regionalisation to the final parameter and DDF values, while Experiment 5 considers a propagation of the two main uncertainty sources interacting together in the final regionalisation of the extremes. The parameter uncertainty is calculated from the number of simulations given in **Table 2** for each experiment, and is illustrated in **Figure 9**; where the upper rows represents the precision (nCI95$_{width}$ [%]), while the lower rows the accuracy (Bias [%]) of estimated parameters in a cross-validation mode. **Figure 10** illustrates the DDF uncertainty at duration levels from 5min up to 7 days for three return periods 1, 10 and 100 years: precision (nCI95$_{width}$ [%]) shown in upper row and accuracy (RMSE [%]) at the lower row. The accuracy of the simulations is compared with the direct regionalisation (i.e. interpolation) of the observed parameter sets (see caption for more details). It is worth mentioning that the difference between the different component simulations (Experiment 1 to 4) is visible only at the precision of the simulations and not at the accuracy. As illustrated by **Figure 9**– lower row and **Figure 10**– lower row, the accuracy at estimating the parameters (Bias [%]) and the DDF values (RMSE [%]) is not changing considerably from one experiment to the other. Also, when comparing the boxplots with the performance obtained from the direct regionalisation (interpolation - shown with the background colours), the same accuracy more or less is observed. Therefore, the analysis will be focused on the variation of precision (nCI95$_{width}$ [%]) according to different sources of uncertainty. Regarding the parameter uncertainty as shown by **Figure 9**, the spatial KED[LS|SS] simulations (Exp. 4) represent the highest source of uncertainties for all the parameters: the nCI95$_{width}$ [%] ranges from 18% for the η parameter, between 40-50% for the two GEV parameters μ and σ, and up to 250% for the θ parameter. For all the parameters, the nCI95$_{width}$ of the KED[LS|SS] simulations are at least 3 times higher than the nCI95$_{width}$ of the other uncertainty sources, concluding that the spatial simulations add to the regionalisation the biggest uncertainty. Second to the KED[LS|SS] simulations, are the resampling of local statistics (Exp. 1) and the OK[SS] simulations (Exp. 3), which seems to produce similar levels of nCI95$_{width}$ for most parameters ranging from 10% for the location - μ, 90% for the θ and only 8% for the η parameter. Only for the scale GEV parameter (σ) is the nCI95$_{width}$ from the local statistics resampling higher (~20%) than the one from OK[SS] (~15%). It is interesting to see, that the obtained nCI95$_{width}$ from the variogram bootstrapping (Exp. 2) are lower than 5% for almost all parameters (exception θ parameter which is lower than 20%). This suggests that the variability between interpolated fields with different variograms is reproducing very similar spatial parameters, even though the variograms differ greatly in terms of nugget, sill and range (see **Figure 5**). The same behaviour is also seen in estimated DDF curves for different return periods (**Figure 10** – upper row), where the variability as exhibited by the variogram bootstrapping (Exp. 2) is very low (less than 10%) compared to the other simulations, and as well constant over the duration levels. On the other hand, the simulations from both local resampling (Exp. 1) and OK[SS] simulations (Exp. 2) exhibit similar patterns of nCI95$_{width}$ for the selected DDFs curves (**Figure 10** – upper row). Unlike the nCI95$_{width}$ exhibited at the parameter simulations, here it is more visible the difference between these two components, as the nCI95% produced by the local resampling (Exp.1) are 1-5% higher than the one produced by the OK[SS] simulations (Exp.3). As seen also in **Figure 10** – upper row, the nCI95$_{width}$ from the KED[LS|SS] (Exp. 4) are the highest compared to the other components, emphasizing that the spatial uncertainty of the KED[LS|SS] is the main source of uncertainty when regionalising the DDF curves. Also, unlike the other types of uncertainties (Exp. 1 to 3), the spatial uncertainty from the KED[LS|SS] depends greatly on the duration level, with nCI9$_{width}$ values of short duration intervals (from 5min up to 2 hours) being considerably higher than the other experiments (reaching on average values of 40%). Moreover, Exp. 4 boxplots are much wider than Exp. 1 to 3, suggesting that the spatial uncertainty is highly dependent on the location. The high uncertainty values in terms of precision for Exp. 4, come with the cost of slightly increased error in RMSE (**Figure 10** -lower row), where the median RMSE values are 1-2% higher than those of the direct regionalisation, but still within the Inter-Quantile-Range (IQR) of the direct regionalisation performance.

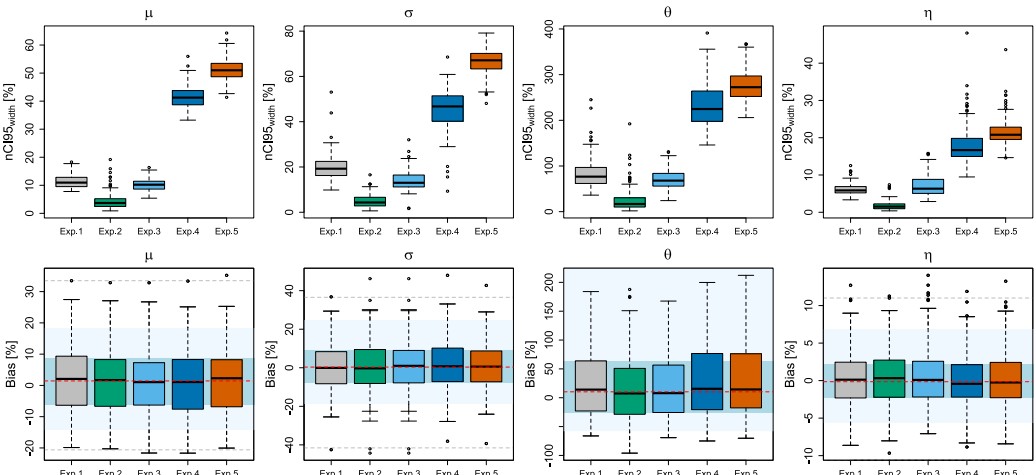

**Figure 9** *The obtained precision (first row - nCI95$_{width}$ [%]) and accuracy (lower row - Bias [%]) from propagating the multiple realisations at different components of the regionalisation procedure to the final parameter sets. The background shades in the lower row illustrate the accuracy of the direct regionalisation (i.e. interpolation) of observed local statistics computed as well in a cross-validation mode, where: red dash indicates the median accuracy over all stations, the blue region the inter-quantile range (IQR) of all stations, the light blue region the 95% and 5% quantiles, and the grey dashed lines the maximum and minimum performance over all stations.*


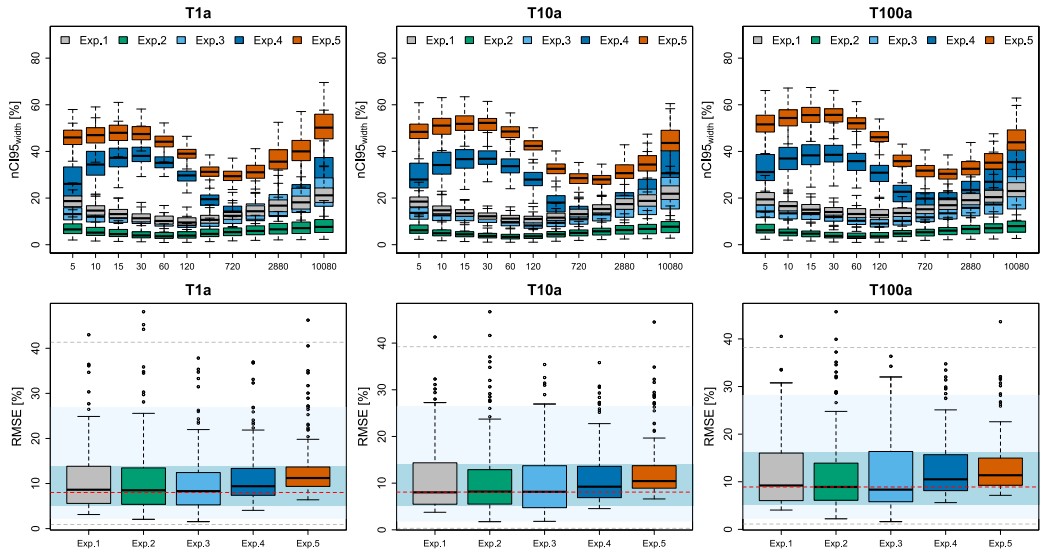


**Figure 10** *The obtained precision (first row - nCI95[%]) and accuracy (lower row - RMSE [%]) from propagating the multiple realisations at each component of the regionalisation procedure to the final DDF values. The background shades in the lower row illustrates the accuracy of the direct regionalisation (i.e. interpolation) of observed local statistics computed as well in a cross-validation mode, where: red dash indicates the median accuracy over all stations, the blue region the inter-quantile range (IQR) of all stations, the light blue region the 95% and 5% quantiles, and the grey dashed lines the maximum and minimum performance over all stations.*





So far, the experiments 1 to 4 considered the propagation of singular uncertainty sources to the final regionalisation of
parameters and DDF curves in Germany. Experiment 5 considers a propagation of the two main uncertainty sources
interacting together in the final regionalisation of the DDF curves. As stated before, the most important sources are; the
local estimation of rainfall extreme statistics, and the spatial uncertainty in regionalisation (KED[LS|SS]). As the
variogram and the external drift is calculated for each local resampling dataset, the uncertainty of variogram and external
drift is already included in the propagation of uncertainty from local resampling to spatial simulations. For each of the
two components, 100 realisations are run, resulting in a total of 10,000 simulations. Overall, the final and total uncertainty
from Exp. 5 follows a similar pattern to the uncertainty from KED[LS|SS] simulations, but due to the local uncertainties,
it manifests higher values of $nCI95_{width}$ and RMSE (as seen in **Figure 9 and 10**). The variation of the total $nCI95_{width}$ for
almost all parameters is 10-20% higher than those of Exp.4, with the GEV parameters reaching values of 50% ($\mu$) to 70%
($\sigma$), the $\theta$ parameter up to 270% and the $\eta$ parameter up to 20%. Consequently, the variation of the total $nCI95_{width}$ over
the duration levels is between 35-50% for return periods 1 and 10 years and between 40-80% for return period of 100
years. As with the KED[LS|SS] simulations (Exp. 4), the durations shorter than 120min and the ones longer than 3 days,
exhibit higher $nCI95_{width}$ values, with the durations from 6 – 48 hours having the highest precision (lowest $nCI95_{width}$
values). Another property seen from experiment 5 is that the variation in space (the wideness of boxplots) is narrower
than in Exp. 4 for most of the durations, suggesting that the final spatial uncertainty is more constant in space (inheriting
a property from local uncertainty – Exp. 1). In term of accuracy, the RMSE [%] has been increased on average with 3%
for 1-year return period, and to 4-5% for 10-100 years return periods, differing slightly from the direct regionalisation
(i.e. interpolation) performance, but still within the Inter-Quantile-Range (IQR) of the direct regionalisation. Since the
median of the simulations from experiments 5 is increasing slightly the RMSE [%] but still within the IQR of the direct
regionalisation, the simulations can be used to quantify the total uncertainty range for the regionalisation of the Depth-
Duration-Frequency Curves. Under this context, the $nCI95_{width}$ [%] values in **Figure 10** can be divided by two, to show
the tolerance range above or below the predicted values at each node from the direct regionalisation. For instance, if at a
specific location, for duration of 5min and return period 100 years, the simulated nCI95[%] is 40%, which means that the
regionalised rainfall depth at this location is varying with $\mp$20% of its mean value.
A parabolic relationship is visible for experiments 1-3, with lower $nCI95_{width}$ values at the mid-duration levels (1 and 2
hours) and increasing values at lower and longer durations. This behaviour is attributed to the Koutsoyiannis framework
for generalising the intensities over all durations by the two parameters $\theta$ and $\eta$. A particular behaviour is the variation
of the $nCI95_{width}$ over the DDF values from the KED[LS|SS] simulations (Exp. 4), which is inherited as well at the final
uncertainty computation (Exp. 5). The behaviour exhibited by KED[LS|SS] simulations does not follow a parabolic
function as in Exp. 1, Exp. 2 and Exp. 3, but more a sinusoidal one. This can be attributed to two main reasons: 1. The
effect of the Koutsoyiannis parameters on different durations, and 2. The spatial simulations of the SGS algorithm
following the transformation to normality.
**Figure 11** – upper row illustrates the observed empirical and simulated CDF from Exp. 4 for each parameter extracted
from the LS dataset. Overall the simulated CDFs agree well with the observed CDFs, however the tails might diverge
slightly. This is particularly true for the lower tail of the $\theta$ and $\eta$ parameters, and the upper tail of the $\sigma$ parameter. This
occurs as the transformation is done on a continuous CDF, a GNO is first fitted to the data and based on the GNO-CDF
the transformation is performed. Nevertheless, this is not negative, as like this, values outside the observed range are
simulated, and hence higher or lower values can be simulated as well. As stated in (Marra et al., 2019b), the rainfall
stations will not capture the maximum intensities of a storm, and thus is almost certain that they don't represent the high
possible intensities. Therefore, generating higher or lower parameter values than observed is crucial in the generation of
stochastic simulations. **Figure 11** – lower row illustrates the correlation between pairs of LS parameters (shown in red



dots), and the corresponding correlations obtained from the 100 KED[LS|SS] simulations run in the cross-validation mode.
For the μ-σ pair the observed correlation is well captured as it coincides with the median of the simulations. To a certain
degree, this is also true for the θ-η pair. The main differences are in the relationships between the GEV and Koutsoyiannis
parameters, where the simulated correlation is much higher than observed. In particular the correlation between μ, σ, and
θ are higher than the correlation between μ, σ, and η. This explains why the precision of the KED[LS|SS] has a sinusoidal
behaviour. The fluctuation of the θ parameter is affecting the uncertainty of the short durations (mainly from 5 to 60min),
while the fluctuation of the η parameter affects the uncertainty at short (5-30min) and very long durations (12hours to 7
days). Since the θ parameter is highly correlated with the μ and σ parameters, its fluctuations will result in a smaller
uncertainty than the η fluctuations, resulting in a slight increase of precision between the duration of 5-30mins.

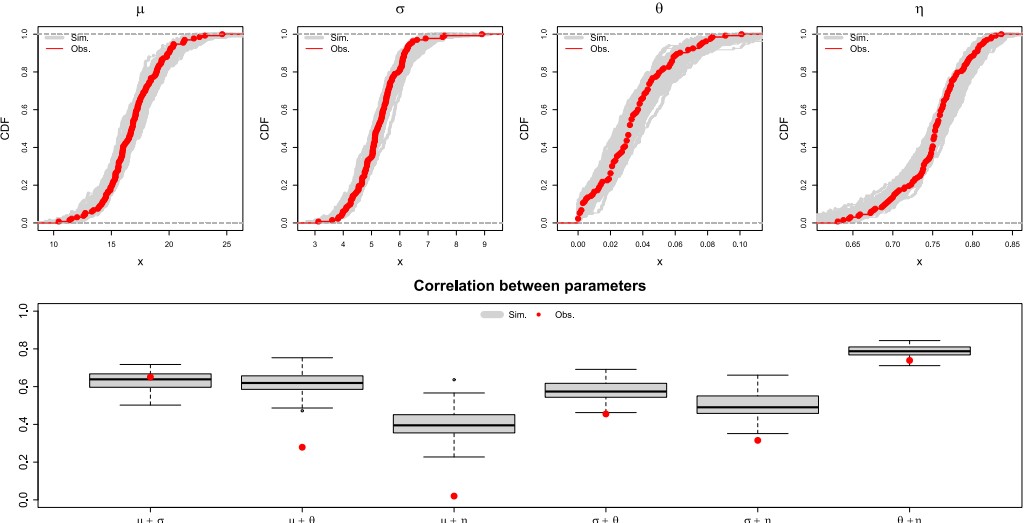

***Figure 11*** *upper row - empirical CDF simulated from Exp.5 (in grey) and from observed paramter values (in red) over*
*the 133 locations; lower row – observed correlations calculated in space between pairs of LS parameters (shown in red*
*dots) and the respective correlations from 100 KED[LS|SS] simulations (shown in the grey boxplots).*

In KOSTRA2010R, which provides design storms for Germany, no objective uncertainty analysis was performed to give
the confidence intervals between 10-20% and hence should not be directly compared with the objective uncertainty esti-
mation performed here. The total uncertainty considered here (from Exp. 5) depends not only on the return period, but as
well on the duration level. The results from **Figure 10** can be used to determine the tolerance above (+) and below (-) the
median for the 95% confidence level. This will result in a median uncertainty range from ±15-25% for low return periods
(lower than 10 years), and from ±20-40% for high return periods (higher than 10 years). Moreover, the short durations
(5min to 2 hours) are in general 20-30% more uncertain than the longer durations (6hours – 1 day). The behaviour exhib-
ited here is in accordance also with other studies (for instance Marra et al., 2017) where the shorter duration intervals are
more uncertain than the ones of 1 day. In this section we compare the uncertainty estimation from two experiments 4 and
5, to see how they distinguish from one another. Uncertainty from experiment 1 is left outside, not only to keep the
graphics simple for visualization, but also because it is much narrower than for the other 2 experiments and it is enclosed
in Exp. 5. Examples of Depth-Duration-Frequency Curves and tolerance ranges for three stations and three returns periods
(T=1, 10 and 100 years) are illustrated in **Figure 12** for three methods: only spatial KED[LS|SS] simulations (from Exp.





4) in blue, local and spatial simulations (from Exp. 5) in orange, and local derived DDF curves in dashed black line. Note
that the results shown here are also obtained in cross-validation mode, which of course overestimate the overall uncer-
tainty at these locations. The first station KO00830 is located in Oberstdorf (a town in the Allgäu Alps of Germany), the
second KO000490 in Soltau Lower Saxony, and the third KO00550 in Emmendingen in the Black Forest region. These
three stations were selected as representative of different regions and behaviours. Over all the stations, the tolerance range
computed by the two experiments are wider at short duration intervals. This is true for all return periods, but the tolerance
ranges get wider with increasing return period. As seen from– first row, the expected rainfall depth in the German Alps is
much higher than the two others, followed by the station in Soltau and the one in the Black Forest. Because of the low
station density in the Alp region, the tolerance range is bigger than in other locations. Overall the two products are similar
with each other, with the main difference present mainly at the durations from 6 to 12 hours, where Exp. 5 exhibits wider
tolerance ranges. Regarding the median estimation of DDF from both experiments, the main difference is seen in the Alps,
where the Exp. 5 agrees better with the observed values. Lastly, we recommend quantifying the uncertainty based on Exp.
5, since the tolerance ranges are better representing the duration levels from 6-12 hours and its median is matching better
with the observation.

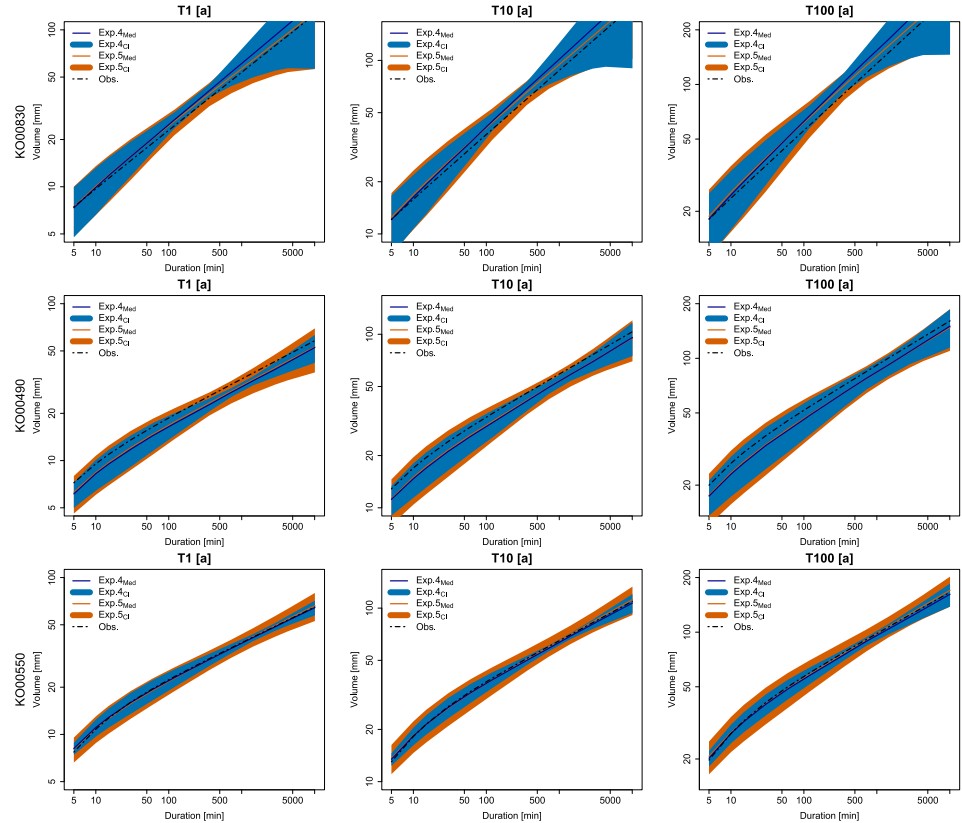


***Figure 12*** *Examples of DDF estimates from observed data and predicted by simulations of Exp. 4 and 5 in a cross-validation mode: as median over all simulations and as 95% tolerance ranges from all simulations: upper row for return period T=1years, middle row for T=10years and lower row for return period T=100years. Three stations are shown here: K000830 located in the German-Alps, KO00490 location in Lower Saxony, and KO00550 located in Black Forest.*



### 4.3 Spatial structure of uncertainty for whole Germany

Spatial maps of precision were generated for three experiments (Exp. 1, 4 and 5), by using the whole dataset, in order to investigate the spatial distribution of the precision when generating the DDFs curves for Germany. The precision in terms of nCI95$_{width}$ [%] for the 4 parameters describing the extreme value statistics are given in **Figure 13.** It can be seen that the different sources of uncertainties exhibit different precision over Germany. For instance, a propagation of the local uncertainty (Exp. 1 showed at the first row), is causing less precision at observed locations (shown in black) than at unobserved location. This is because, the resampling of the target network (LS) proves more uncertainty than resampling the external drift network (SS). Therefore, uncertainty estimated from Exp. 1 is not enough to capture the spatial structure of the uncertainties. On the other hand, Exp. 4 shows a clear spatial structure for uncertainty (mainly for three parameters σ, θ, and η) with the North-West and South of Germany having higher uncertainty ranges. This follows the precipitation regime and the station density in Germany; the South parts records higher precipitation amounts because of the German Alps (so it is a region with clearly different behaviour than the rest of Germany), while the North-West has a lower station density for both the LS and SS datasets in comparison with the rest of Germany. The uncertainty range at two parameters μ and σ is increasing with 30-40% for whole Germany when combining the local with spatial uncertainty (Exp. 5) in comparison with only spatial uncertainty (Exp. 4). The uncertainty at the parameters θ and η remains more or less at similar levels, with similar spatial patterns. Thus, including the local uncertainty mainly influences the parameters of the GEV distributions. It is interesting to see in Exp. 5, that at the location of the long stations (shown in black squares), the uncertainty of the parameters μ and σ is much lower than for the rest of the regions. This is an expected behaviour, as observed locations should be more certain than unobserved ones, and as the station density decreases, so increases the uncertainty. This behaviour, not seen in other experiments, seems to be captured quite well by Exp. 5. This is particularly true for the GEV parameters, while the Koutsoyiannis parameters show an additional spatial variability of uncertainty that follows the main elevation features in Germany (represented by the external drift): with North-West and South Germany having higher uncertainty ranges. Another interesting point is the high uncertainty associated at the σ parameter by Exp. 5 at Münster city (shown in a red circle) which is as well visible at Exp. 1. The high uncertainty of the scale parameters comes mainly from the local resampling bootstrap. As discuss in Shehu. al (2022) a very rare extreme event has been recorded in 2014 in Münster, which affects the extreme value analysis considerably. Thus, the integration of the local uncertainty becomes mandatory to estimate the uncertainty when including these rare events (with a very high return period) in the estimation of DDF curves for design purposes.

**Figure 14** illustrates the spatial distribution of uncertainty (computed here in term of precision nCI95$_{width}$ [%]) for the durations 5min, 1hour, and 1 day and return period of 100 years: upper row - only from local uncertainty (Exp. 1), second row – only from spatial uncertainty (Exp. 4) and lower row – from both local and spatial uncertainty (Exp. 5). The uncertainty ranges exhibited by Exp. 1 (only considering the local uncertainty) are very similar throughout all three durations and maintain similar spatial structure as with the parameter uncertainty in **Figure 13**. Here, the difference between observed and unobserved locations is small and, following the parameter precision, the observed locations have higher uncertainty that the unobserved ones (on average 15-20% higher nCI95$_{width}$ values). In Exp. 4 there is a clear difference between the uncertainties of different durations, where the uncertainty of very short and very long durations (5min and 1day) are governed by the spatial structure of θ and η parameters. The uncertainty of 1-hour durations are more or less uniformly distributed, but with the North-West region exhibiting higher uncertainties than the rest of Germany. At Exp. 5 the uncertainty for 5min durations has been increased considerably when including the local uncertainty (from 20-55% in Exp. 4 to 80-100%). The uncertainty of 1-hour durations exhibits similar patterns but is increased slightly from 45% to 55% at Exp. 5. For 1-day duration, the uncertainty ranges are as well increased by Exp. 5, with values higher at the southern part of Germany (where the German Alps are located) and at the northern part of Germany near to the North





Sea. The extreme event at Münster, influences the uncertainty of all durations but has a higher impact of short durations.
Based on such propagation of uncertainty, tolerance ranges between ∓30-60% should be expected in Germany for 5min
duration intervals, ∓15-45% for 1-hour durations and ∓20-50% for 1-day durations. Overall, the combination of local
resampling with geostatistical spatial simulations provides the best method for the assessment of uncertainty in
regionalisation DDF curves in Germany. First, and most importantly, the precision of these curves is higher at the location
of long stations, and decreases in ungauged locations according to the distance from the long observations and the density
of the observations in the vicinity.

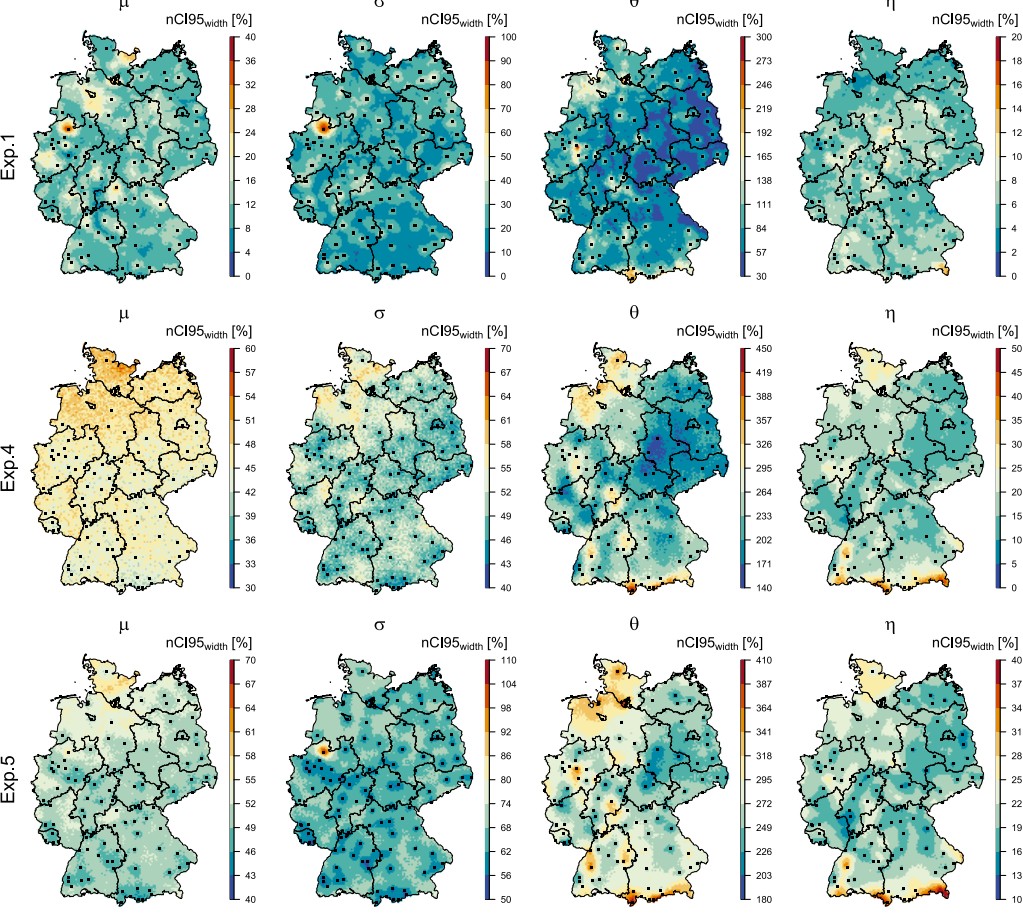


***Figure 13*** *The precision (nCI95[%]) in estimating the 4 parameters for the whole Germany will all available data for two experiments: upper row – results obtained from the propagation of 100 local resampled data to the final regionalisation (Exp. 1), middle row - results obtained from 100 spatial simulations of KED[LS|SS] (Exp. 4), lower row – results obtained from 10,000 local resampling and spatial simulations of KED[LS|SS] (Exp. 5). The black squares indicate the locations of LS, while the black lines illustrate the boundaries of German Federal states. Note that the ranges for the legend colours are changing for each experiment in order to emphasize the spatial structure of each experiment.*



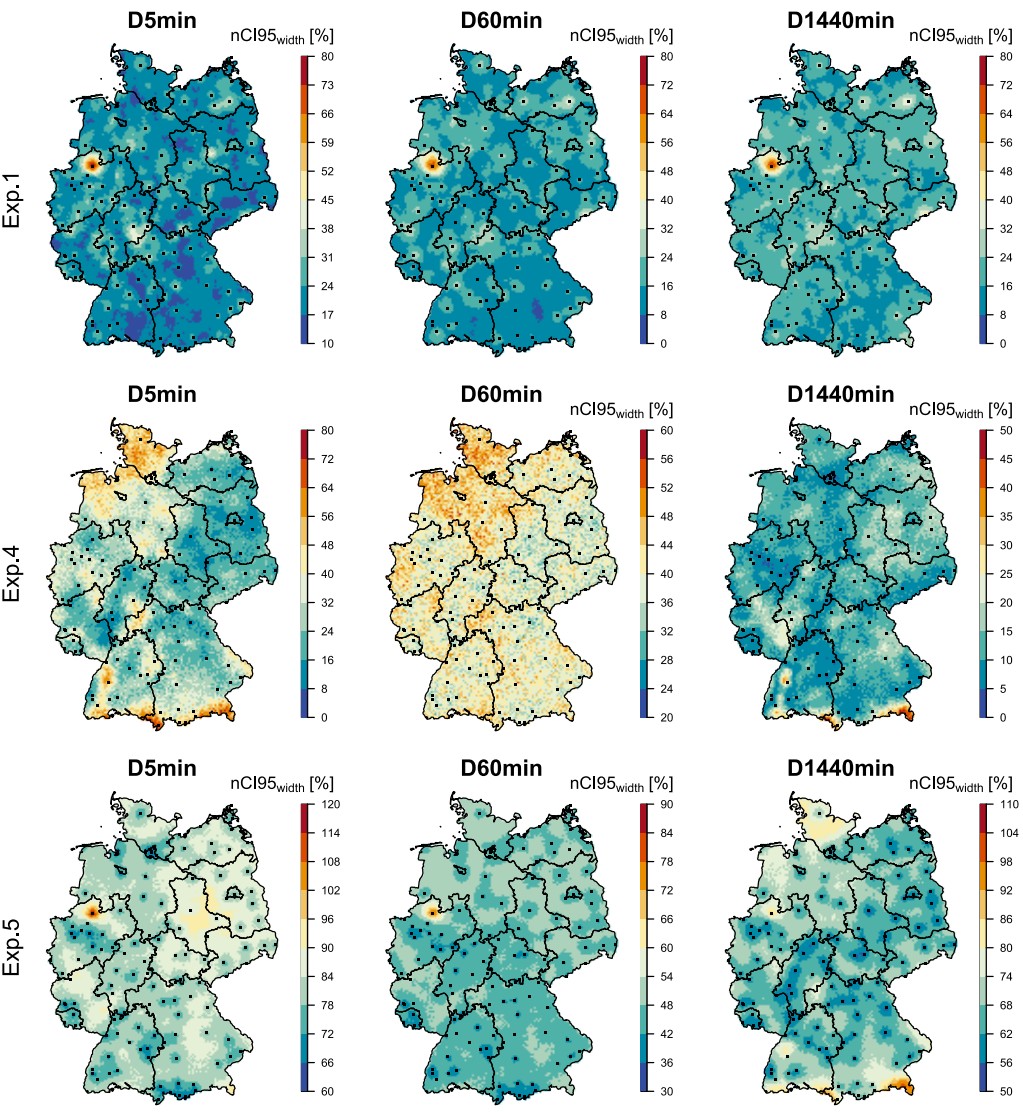


***Figure 14*** *The precision (nCI95[%]) in estimation rainfall depth at different durations and 100 year return period for whole Germany with all available data for three experiments: upper row – results obtained from the propagation of 100 local resampled data to the final regionalisation (Exp. 1), middle row –results obtained from 100 spatial simulations of KED[LS|SS] (Exp. 4), lower row – results obtained from 10,000 local resampling and spatial simulations of KED[LS|SS] (Exp. 5). The black squares indicate the locations of LS, while the black lines illustrate the boundaries of German Federal states. Note that the ranges for the legend colours are changing for each experiment in order to emphasize the spatial structure of each experiment.*



## 5. Conclusion and Outlook

In Shehu et al. (2022), a regionalisation based on external drift kriging was employed to calculate Depth-Duration-Frequency (DDF) curves in Germany. Based on these results, an uncertainty analysis was conducted here to estimate the precision of the obtained regionalised DDF curves in Germany. For this purpose, many simulations were performed at the main components of the regionalisation procedure: local estimation of the extreme statistics (by non-parametric bootstrapping), spatial dependency (by variogram bootstrapping) of short and long stations statistics, the external drift information (by Sequential Gaussian Simulations) and the interpolation (also with Sequential Gaussian Simulations). Four different experiments were run in order to estimate how the uncertainty at each component propagates to the final regionalisation of the DDF curves, and a last experiment was performed by combing the uncertainty of the two main components in order to assess the total uncertainty. The uncertainty, in terms of precision, was evaluated at each long station location (on a cross-validation mode) based on the obtained 95% confidence interval from different simulations. The conclusions from this investigation are summarised below:

- A comparison with Simulated Annealing showed that the SGS is better suitable for the study at hand, as it shows higher accuracy by capturing better the inter-relationship between the parameters (despite of the data transformation). Further works may include a new SA algorithm that models of the 4 parameters together in space in order to keep the inter-relationship between them. A future improved SA algorithm may have the potential to decrease the overall uncertainty ranges of DDF curves further on.

- The uncertainty from the variograms, that describes the spatial dependencies within the short and long observation datasets, does not seem to influence much the final regionalisation of parameters and hence the estimation of the DDF curves. Therefore, it was neglected for the total uncertainty propagation. On the other hand, the uncertainty from the regionalisation of the long observations is the biggest source of uncertainty, followed up by the local estimation of extremes and by the drift estimation from short observation. A bootstrapping technique that combines the local estimation of extremes together with different spatial simulations of the long observations, provided the highest uncertainty and was used to quantify the total uncertainty.

- The total uncertainty obtained here follows mainly the behaviour of the spatial uncertainty, but is slightly higher, as it is influenced by the local uncertainty. However, unlike the spatial uncertainty, the total uncertainty is influenced by the very rare extreme events, and considers them as well for the computation of tolerance ranges. Moreover, by combining local resampling with spatial simulations, the modelled uncertainty exhibits a valid behaviour: at observed locations the precision is higher, and it decreases at unobserved locations according to the distance from the observed, and the density of the observed locations in the vicinity. For very short and very long durations, uncertainty ranges are also dependent on different climatological regions in Germany.

- From 10,000 simulation, it is concluded that the durations shorter than 2 hours exhibit a larger uncertainty that longer durations, which of course is increasing with the return period considered. Depending on the location and duration, tolerance ranges from ±10-30% for low return periods (lower than 10 years), and from ±15-60% for high return periods (higher than 10 years) should be expected.

- For the proposed methodology, the uncertainty variation in space (for most locations) seems to be smaller (~10-20%) than the variation across different durations (up to 30%). On the other hand, the uncertainty variation due to the return periods is low, approximately 5 to 10%. The only exception is at Münster, where a very rare extreme events has been observed and causes high uncertainty ranges for the extreme values in the vicinity. Events such at the one in Münster, influence the DDF curves considerably, and hence more research should be done in order to investigate how to treat them when the focus is on DDF curves for return periods up to 100 years.



Overall, the combination of local resampling with geostatistical spatial simulations provides a very suitable method for
the assessment of uncertainty in regionalisation DDF curves. As shown here, considering only local resampling for the
sample variability will underestimate the total uncertainty especially at very short duration interval and high return periods.
Therefore, it becomes crucial to include as well spatial simulations for the computation of uncertainties. In this study, the
extreme value analysis based on GEV was investigated, nevertheless it would be interesting to see if a meta-statistical
approach, as proposed by Marra et al. (2019a), can result in narrower tolerance ranges while keeping a higher accuracy.
So far, only the sample and spatial variability were included for the estimation of the uncertainties. Future works may as
well include non-stationarity due to climate change, and the change of uncertainty patterns in the future.

### 6. Data Availability

The daily and the short sub-daily network are made publicly available by the German Weather Service (DWD) and can
be accessed at https://opendata.dwd.de/climate_environment/CDC/. All R-codes can be provided by the corresponding
authors upon request.

### 7. Authors Contribution

Supervision and funding for this research were acquired by UH, the study conception, design and methodology were
performed by both authors, while the software, data collection, derivation and interpretation of results were handled
mainly by BS. BS prepared the original draft, which was revised by UH.

### 8. Competing Interest

The authors declare that they have no conflict of interest.

### 9. Funding

This research was funded by the German Ministry of Agriculture and Environment Mecklenburg-Vorpommern and the
Federal State Funding Programme "Water, Soil and Waste".

### 10. Acknowledgements

The results presented in this study are part of the research project "Investigating Different Methods for Revising and
Updating the Heavy Rainfall Statistics in Germany (MUNSTAR)", funded by the German Ministry of Agriculture and
Environment Mecklenburg-Vorpommern and the Federal State Funding Programme "Water, Soil and Waste" who are
gratefully acknowledged. We are also thankful for the provision and right to use the data from the German National
Weather Service (Deutscher Wetterdienst DWD), more specific Thomas Deutschländer and Thomas Junghänel, and to
Winfried Willems from the Institute of Hydrology, Applied Water Resources and Geoinformatics (IAWG) for their
contribution in the local extreme value analysis.

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
