# Peer review of "Uncertainty Estimation of Regionalised Depth-Duration-Frequency Curves in Germany"

_Hydrology and Earth System Sciences, 2022_

## Referee Comment (RC1)

**General comments**

This works deals with the characterization of uncertainty in regionalized design rainfall curves, (so-called depth-duration-frequency (DDF) curves) constructed for the entire Germany in an accompanying work by a larger research group, also involving the authors of the present paper. The former paper is currently under review for the same journal (Shehu et al., 2022). A lot of choices made in the current paper are dictated by the way the subject was approached in the first paper and, to a certain extent, constrained by it. Although the first paper is currently still under review, I find the approach carried out in it reasonable and I will limit my review to how uncertainty in these choices was handled, not intending to extend my review to both works.

This paper focuses on identifying and exploring various components of uncertainty present in the regionalized design rainfall curves, with a particular emphasis on the spatial structure of uncertainty. This is an important, challenging and under-explored research subject and the efforts that the authors have undertaken to approach it are considerable and worth of appraisal.

The paper is very well-written, the results are discussed thoroughly and the authors strive to provide insights into a challenging subject. A great amount of computational work is obviously involved. I wish to congratulate the authors for these efforts and I suggest publication in HESS upon some revisions/clarifications that can be addressed by minor alternations/additions to the manuscript. I discuss these below along with some suggestions from improvements that might be found useful.

**Specific comments**

-On uncertainty of DDF parameters (referred to as 'local' uncertainty)

By the investigation carried out in the first paper, the authors conclude on using a common, fixed value of the shape parameter, based on the literature for return periods up to 100 years (in particular, following Koutsoyiannis' work (2004)). Using a common value of the shape parameter is a reasonable choice given the high uncertainty involved in its estimation from single stations. However, fixing the value of the shape parameter based on the literature, instead of estimating it, is not devoid of uncertainty either. It may not be straightforward to assess the uncertainty of a fixed parameter, but on the other hand, neglecting the shape parameter from the uncertainty assessment is a limitation that I think should be discussed. Perhaps the authors could add a brief discussion of the uncertainty associated with the shape parameter, and comment on its expected impact on local uncertainty, which presumably would be exacerbated if this was also accounted for.

-On regionalization uncertainty

The quantification of the spatial uncertainty presents the greatest challenge and also, the major contribution of the work. The authors could elaborate more on the reasons why they employ

spatial simulations for assessing regionalization uncertainty, since this is the only component of the uncertainty analysis for which simulation is employed. For instance, Lines 87-90 could be better explained; the reasons that the authors do not employ kriging for spatial uncertainty are less clear to someone not familiar with the provided geostatistical literature. In particular, given that they consider kriging variance to be a measure of uncertainty for the unobserved locations (Line 88), why is it later claimed to be only efficient for the local (I understand this as 'at-station') uncertainty and not the spatial one? The question also arises since the Sequential Gaussian Simulation that the authors finally choose for the assessment of spatial uncertainty is based on the kriging mean and variance. Do the authors use spatial simulations mainly because they seek to produce richer spatial patterns than the ones obtained by a kriging interpolation? A brief explanation of the terms 'local' and 'spatial uncertainty' would also be helpful to the reader at this point (a definition is provided later on, in Section 3).

My understanding is that in the parts of Experiments 3-5 that involve spatial simulations, the authors produce prediction intervals, which are subtly different than the confidence intervals obtained by the bootstrapping procedure followed for the assessment of local and variogram uncertainty. Could the authors comment on this difference and clarify, if appropriate?

Line 306-307: Could the authors explain the rationale for performing the conditional simulation, i.e. maintaining in the simulation a set of observed values for the long stations, and also, explain how this set is chosen?

-On the comparison among different components of uncertainty

The authors find that 'the spatial simulations add to the regionalisation the biggest uncertainty', and based on that conclude that 'the spatial uncertainty is the main source of uncertainty when regionalising the DDF curves'. Although this finding is reasonable and supported by the results presented, I also deem the estimation method important in the context of the inter-comparison of the different uncertainty intervals. Perhaps, this might not be the determining factor, but still could the observed difference between the spatial uncertainty and the other sources be, in part, due to the fact that (spatial) simulation produces wider intervals than bootstrapping?

-Conclusions

From the large experience gained through this work, the authors could perhaps comment on whether/how the observed uncertainty could be constrained by targeted data collection. For instance, what do they think should be prioritized in terms of data collection, and could there be an added benefit from incorporating gridded precipitation/satellite products in their framework?

**Technical/minor comments**

Line 59: The phrase 'non-representativeness of point-measures' appears quite vague; I suggest adding a short explanation of what is meant by it.

Line 188: Typo inside the parenthesis, μ should be replaced with η.

Line 196: Do the authors mean a 5 km x 5 km grid here?

Line 201: I suggest '…have an inadequate length for' instead of '…too little observation years for'.

Line 208: Probably 'deviate' is meant instead of 'denote' here.

Line 222: 'method' instead of 'moment'.

Line 297: Could the authors justify the choice of 133 stations here?

Line 365: 'fixing' instead of 'fixating'.

Line 508-509: Could the authors clarify what is implied here for the behavior of the Koutsoyiannis' parameters? Does the explanation provided in Lines 529-530 apply here as well?

Figure 12: Please check subplots' numbering and consider replacing 'volume' with 'depth'.

Figure 13: 'with' instead of 'will'.

Please check numbering of the sections; there are two sections '2.1'.

With best regards,

Theano Iliopoulou

**References**

Shehu, B., Willems, W., Stockel, H., Thiele, L.-B. and Haberlandt, U.: Regionalisation of Rainfall Depth-Duration-Frequency curves784 in Germany, Hydrol. Earth Syst. Sci., [preprint], 2022.

Koutsoyiannis, D.: Statistics of extremes and estimation of extreme rainfall: II. Empirical investigation of long rainfall records, Hydrol. Sci. J., 49(4), 591–610, doi:10.1623/hysj.49.4.591.54424, 2004b

---

## Author Comment (AC1)

*__Response to Referee #1,__*

Dear Theano Iliopoulou,

Thank you for taking the time to evaluate our work, and as well for your comments that will for sure improve our manuscript. Please find below our answers to the issues you have raised.

- **Comment1**: On uncertainty of DDF parameters (referred to as 'local' uncertainty)
  By the investigation carried out in the first paper, the authors conclude on using a common, fixed value of the shape parameter, based on the literature for return periods up to 100 years (in particular, following Koutsoyiannis' work (2004)). Using a common value of the shape parameter is a reasonable choice given the high uncertainty involved in its estimation from single stations. However, fixing the value of the shape parameter based on the literature, instead of estimating it, is not devoid of uncertainty either. It may not be straightforward to assess the uncertainty of a fixed parameter, but on the other hand, neglecting the shape parameter from the uncertainty assessment is a limitation that I think should be discussed. Perhaps the authors could add a brief discussion of the uncertainty associated with the shape parameter, and comment on its expected impact on local uncertainty, which presumably would be exacerbated if this was also accounted for.

**Response:** We understand your concern regarding the fixed shape parameter to the value of 0.1. Please note that the value 0.1 was not purely theoretical (following the Koutsoyiannis work in 2004). We calculated the shape parameter for all the long stations (with long observations) and the shape parameter had a median close to 0.1. Also, Ulrich et al. 2021 computed the GEV shape parameter of annual extremes in Germany based the Koutsoyiannis mathematical framework as 0.11. Thus, the decision to fix the shape parameter at 0.1 was not trivial and not just based on the Koutsoyiannis work, but of previous analysis that we have conducted. This is something that we have tried to make clearer in the updated version of the first paper (Shehu et al. 2022). Nevertheless, thank you for pointing that out, and we will add a better clarification for this in this manuscript.

Also, we will add a description about the uncertainty of a free shape parameter (estimated independently for each station) to the local uncertainty. Please keep in mind that in the previous work of Shehu et al. 2022, we have investigated which method is more precise for local estimation of the rainfall extremes: keeping the shape parameter free or fixed. In our previous work, it was visible that the $nCI95_{width}$ was much higher for the free shape parameter than for the fixed one. Below you will see the normalized 95% Confidence Interval width ($nCI95_{width}$) obtained by local bootstrapping of annual maximum series where the shape parameter was kept constant as calculated for each station (in the Figure 1 donated as KO.FREE) and fixed to 0.1 for all stations (as implemented in this manuscript, denoted with KO.FIX). These confidence intervals widths were computed for each of the 133 long observations, for each duration and for each return period shown in the legend. It is clear that for return period higher than 20 years, the uncertainty from a free shape parameter is much higher than the uncertainty from keeping the shape parameter fixed at 0.1, which will cause the interpolation of extreme rainfall to be less certain (and with higher uncertainty ranges). Please note that the following Figure 1 will be not added at the manuscript, but serves only for this review in order to explain the difference between the two.

[Figure]

Figure 1 A direct comparison of the normalized confidence interval width for two cases of the local sample uncertainty: case 1 – KO.FIX where the GEV-shape parameter is fixed as 0.1 (x-axis) and case 2 KO-FREE where the GEV shape parameter is computed independently for each station (y-axis). The comparison is done for each station, duration and return period (shown in different colors in the legend).

- **Comment 2:** On regionalization uncertainty

The quantification of the spatial uncertainty presents the greatest challenge and also, the major contribution of the work. The authors could elaborate more on the reasons why they employ spatial simulations for assessing regionalization uncertainty, since this is the only component of the uncertainty analysis for which simulation is employed. For instance, Lines 87-90 could be better explained; the reasons that the authors do not employ kriging for spatial uncertainty are less clear to someone not familiar with the provided geostatistical literature. In particular, given that they consider kriging variance to be a measure of uncertainty for the unobserved locations (Line 88), why is it later claimed to be only efficient for the local (I understand this as 'at-station') uncertainty and not the spatial one? The question also arises since the Sequential Gaussian Simulation that the authors finally choose for the assessment of spatial uncertainty is based on the kriging mean and variance. Do the authors use spatial simulations mainly because they seek to produce richer spatial patterns than the ones obtained by a kriging interpolation? A brief explanation of the terms 'local' and 'spatial uncertainty' would also be helpful to the reader at this point (a definition is provided later on, in Section 3).

**Response**: Following are the main reasons that justify our choice in employing simulations instead of using the kriging variance. We will update the text and make sure the reasons are clearly stated.

1. As you have already mention, the kriging interpolation is causing smoother spatial patterns, and with the simulations we are trying to produce rich spatial patterns (Deutsch & Journel, 1998).
2. The kriging variance depends only on the data configurations (how many stations there are and where they are in relation to each other) but they are independent of the observed values. With the simulations we try to compute uncertainty also in terms of the observed values (in addition to data configuration) (Deutsch & Journel, 1998).
3. Spatial simulations provide a measure of uncertainty about the un- sampled values taken altogether in space rather than one by one. They can provide information regarding joint-spatial uncertainty, unlike kriging variance which provides information at one location at a time conditioned to other observations in the vicinity. Thus, under the context of the kriging systems, local refers to the uncertainty prevailing at one location, while spatial refers to the joint uncertainty prevailing at multiple locations (Deutsch & Journel, 1998).  We will clarify these two terms in the lines 87-90.

4. Lastly and more importantly, to estimate confidence intervals as employed here, we should assume that the kriging prediction errors are normally distributed in order to calculate the confidence intervals from the kriging variance. But please keep in mind that we are interpolating parameters that have a specific relationship with each other. If we want to compute the 95% confidence interval for the depth-duration-frequency curves, it is not possible to combine the quantile values of each parameter, as they may not correspond to each other. That is why for this case simulation is much better than the kriging interpolation.

My understanding is that in the parts of Experiments 3-5 that involve spatial simulations, the authors produce prediction intervals, which are subtly different than the confidence intervals obtained by the bootstrapping procedure followed for the assessment of local and variogram uncertainty. Could the authors comment on this difference and clarify, if appropriate?

**Response**: Regarding experiments 3-5, I'm not quite sure if I understand your concern correctly. I suppose you mean why the confidence intervals obtained by the Experiments 4 and 5, have much higher values and exhibit a completely different shape across the durations, than the Experiment 1, 2 and 3 (related to Figure 10 – upper row). We have tried to explain this behavior at the lines 507-532. Additionally, please refer to our reply on the comment 3.

Line 306-307: Could the authors explain the rationale for performing the conditional simulation, i.e., maintaining in the simulation a set of observed values for the long stations, and also, explain how this set is chosen?

**Response**: Yes, we have performed a conditional simulation, which means that for known locations (where we have observations), we are not simulating as we have already observed those pixels. By simulated annealing the pixels are conditioned to the exact observations, while in sequential gaussian simulations, the pixels are allowed to vary from the exact observations as per the nugget coefficient of the variogram. We will add an explanation about this in the manuscript. Regarding the reason why we chosen the long station as a set in explain in the Shehu et al. 2022 manuscript, but we will add an explanation in this manuscript as well. Long stations are more reliable than short stations when estimating extreme rainfall, that is why there are recognized as the primary set of data: they are used as the ground truth (for the cross-validation), and as well the main input for the regionalization (short series are used as an external drift to the long series interpolation). In this study we have only 133 stations, because those were the stations available at the time being from German Weather Service with 1min temporal resolution for observation length more than 40 years.

- **Comment 3:** Comparison among different components of uncertainty
The authors find that 'the spatial simulations add to the regionalization the biggest uncertainty', and based on that conclude that 'the spatial uncertainty is the main source of uncertainty when regionalizing the DDF curves'. Although this finding is reasonable and supported by the results presented, I also deem the estimation method important in the context of the inter-comparison of the different uncertainty intervals. Perhaps, this might not be the determining factor, but still could the observed difference between the spatial uncertainty and the other sources be, in part, due to the fact that (spatial) simulation produces wider intervals than bootstrapping?

**Response**: That is an interesting point and yes, I agree with you, the spatial simulations are in fact producing wider intervals than the local bootstrapping. This can be also seen by comparing the variance of the local bootstrapping with the variance in space of the observations. So, the parameters are varying more in space, and that is why when sampling from space (spatial simulations) the prediction intervals are higher than for the bootstrapping case. We will add a small discuss about this in the manuscript.

- **Comment 4**: Conclusion

From the large experience gained through this work, the authors could perhaps comment on whether/how the observed uncertainty could be constrained by targeted data collection. For instance, what do they think should be prioritized in terms of data collection, and could there be an added benefit from incorporating gridded precipitation/satellite products in their framework?

**Response**: Unfortunately, we have not yet investigated how to target the data collection in order to reduce the uncertainty. This is definitely something that we would like to try in the future. It can be that using a gridded product (like radar or satellite products) may be useful in reducing the uncertainty in the spatial simulations. Nevertheless, this might be problematic as the radar or satellite products might filter inaccurate readings when searched for extremes.

- **Comment 5**: Technical/minor comments

The phrase 'non-representativeness of point-measures' appears quite vague; I suggest adding a short explanation of what is meant by it.

Lines 58-60 has been updated to: "In this paper, the focus is on developing a framework that accounts for uncertainties due to short observation lengths and non-representativeness of point measurements for spatial dependencies of extremes."

Line 188: Typo inside the parenthesis, $\mu$ should be replaced with $\eta$.

Thank you for noticing it. The symbol was changed correctly.

Line 196: Do the authors mean a 5 km x 5 km grid here?

Yes, that is exactly what we mean. We have changed it to 5x5km.

Line 201: I suggest '...have an inadequate length for' instead of '...too little observation years for'

We accept your suggestion and we have changed it accordingly.

Line 208: Probably 'deviate' is meant instead of 'denote' here.

Yes, this is true and we have changed it accordingly.

Line 222: 'method' instead of 'moment'.

Thank you for noticing it. We have change it accordingly.

Line 297: Could the authors justify the choice of 133 stations here?

Only 133 values from all the stations were sampled here, to address the uncertainty in computing the variogram from a small dataset that corresponds with the number of the long stations that were used to compute the variogram for the KED interpolation. We will add a short explanation in the text regarding this.

Line 365: 'fixing' instead of 'fixating'

Thank you for noticing it. We have change it accordingly.

Line 508-509: Could the authors clarify what is implied here for the behavior of the Koutsoyiannis' parameters? Does the explanation provided in Lines 529-530 apply here as well?

In lines 508-509, "this behavior of the Koutsoyiannis parameters" is referring to why for experiments 1 to 3, the confidence interval width across the durations exhibits a parabolic shape. And yes, the lines 529-530 apply also here.

Figure 12: Please check subplots' numbering and consider replacing 'volume' with 'depth'.

The y-axis labels for the Figure 12 have changed from Volume to Depth. Please see below the updated Figure 12.

[Figure]

*Figure 12 Examples of DDF estimates from observed data and predicted by simulations of Exp. 4 and 5 in a cross-validation mode: as median over all simulations and as 95% tolerance ranges from all simulations: upper row for return period T=1years, middle row for T=10years and lower row for return period T=100years. Three stations are shown here: K000830 located in the German-Alps, KO00490 location in Lower Saxony, and KO00550 located in Black Forest.*

Figure 13: 'with' instead of 'will'

Thank you for noticing it. We have change it accordingly.

Please check numbering of the sections; there are two sections '2.1'.

Thank you for noticing it. We have change it accordingly and checked the rest of the numbering.

With kind regards,

Bora Shehu

References:

Deutsch, C. V. and Journel, A. G.: GSLIB: geostatistical software library and user's guide. Second edition., 1998.

Koutsoyiannis, D.: Statistics of extremes and estimation of extreme rainfall: I. Theoretical investigation, Hydrol. Sci. J., 49(4), 575–590, doi:10.1623/hysj.49.4.575.54430, 2004a.

Koutsoyiannis, D.: Statistics of extremes and estimation of extreme rainfall: II. Empirical investigation of long rainfall records, Hydrol. Sci. J., 49(4), 591–610, doi:10.1623/hysj.49.4.591.54424, 2004b.

Shehu, B., Willems, W., Stockel, H., Thiele, L.-B. and Haberlandt, U.: Regionalisation of Rainfall Depth-Duration-Frequency curves in Germany, Hydrol. Earth Syst. Sci., [preprint], 2022

Ulrich, J., Fauer, F. S. and Rust, H. W.: Modeling seasonal variations of extreme rainfall on different timescales in 774 Germany, Hydrol. Earth Syst. Sci., 25(12), doi:10.5194/hess-25-6133-2021, 2021.

---

## Author Comment (AC2)

***Response to Referee #2,***

Dear Referee,

Thank you for taking the time to evaluate our work and as well for your comments that will for sure improve our manuscript. Please find below our answers (in blue) to the issues you raised (in black):

- **Comment 1:** Short durations have a quite large uncertainty that, in absolute value, may be very relevant for practical applications (e.g., from Fig 12 it can be about +/- 2 or 3 on a mean value of 6 or 7). This can be expected as the rainfall processes that generate short-duration extremes are usually different from those generating hourly/daily maxima. In general, these short-duration events are more difficult to interpolate because are very "local". Do the authors have investigated this aspect and studied how the variogram characteristics (in particular the range) vary with the duration and impact the uncertainty?

   **Response**: Yes, we agree with your point of view. However please note that we haven't interpolated the GEV-parameters or -quantiles for each duration independently. Prior to interpolation, we have used the Koutsoyiannis framework to pool together all durations (from short to very long). Therefore, we haven't investigated the independent interpolation of parameter or quantiles for each duration separately. This would for sure be something interesting to investigate in future works, but on the other hand it might be problematic as quantile crossing may occur. With quantile crossing here is meant when the quantiles of two durations are interpolated independently and consequently the rainfall depth of a longer duration may be smaller than those of a shorter duration.

   In the methodology that we have applied here, the spatial structure of durations shorter than 60min is dictated mainly by the θ parameter, while the structure of durations longer than 2 hour by the η parameter (although please note that the η parameter has still some effect on the short durations). The variograms or these two parameters are shown in Figure 5, where it is visible that the range of the θ parameter is approximately 50km, while that of the η parameter approximately 70-150 km. This suggest that the short duration have shorter range and hence are more local.

   We have performed experiment 2 to investigate how different variogram characteristics are affecting the interpolation and the uncertainty of the parameter θ (please refer to figure 5, 9, and 10 in the manuscript). So, when we perform the interpolation with different variograms that differ in sill, nugget and range, there is not a considerable difference in the interpolation accuracy or precision. The uncertainty variation among the different duration is small, but the short duration levels are still slightly less certain. In the experiments conducted here, we have not varied each of the variogram features (nugget, sill and range) independently, to see their exact influence on the uncertainty. This is something that we will consider for future works.

- **Comment 2:** In fig 10, bottom row, there are several outliers. Do the authors have an interpretation for this behavior (e.g., can be related short time series)? Are these points also "extreme" in terms of parameters (mu, sigma, theta and/or eta) or this behavior emerges only looking at quantiles? Are these stations geographically clustered?

   **Response**: We have identified the reasons why some of the outliers are present, and this behavior emerges actually both from parameter extremes and by looking at the quantiles. For instance, there is the station in Münster City (also mentioned in the manuscript) where a scale parameter outlier is located (this location is also visible from the spatial uncertainty maps where a local high uncertainty is visible). Here a very rare extreme event was observed which is shifting the scale GEV parameter to be particularly different from the stations in the surrounding. This causes as well the RMSE of high return periods (T100a) rainfall depths to be higher in this location. Other examples are for instance

stations whose parameters are considerably different from the neighboring long observations, and hence when simulated they exhibit a high error (both parameters and consequently rainfall depths). This is observed in singular stations in the Black Forest or in the German Alps.

However, on the other side, we have also observed that the locations of rainfall depth outliers do not correspond to the locations of parameter outliers (and vice versa parameter outliers do not always lead to rainfall depth outliers). This suggests that these outliers are also emerging from looking at the quantiles of different simulations. Also, we haven't recognized any regional clusters where several stations in the vicinity are exhibiting outliers in terms of RMSE.

- **Comment 3:** P12 L360 the meaning of "reduction factor" is not clear and the symbol lambda is already used in eqs 7-8. I suggest removing it.

  **Response**: The idea is to start with a high temperature at the beginning, and each iteration a perturbation is accepted, the temperature is reduced by multiplying it with a factor (in our case the factor is 0.1). With decreasing temperature, it is more difficult to accept the perturbation, until the moment the temperature is very low and the image is frozen (the optimization has been reached). We will add a brief explanation to clarify the read and change the lambda symbol.

- **Comment 4:** FIG 14 Please consider using the same color scale for each plot of the same duration to facilitate comparison

  **Response**: I understand your concern; however, we would prefer to leave the figures as they are in order to recognize better the spatial structure of the uncertainty. As the difference between the experiments is quite big (mainly between Exp.1 and Exp.5) the spatial structures will be lost. For instance, in Exp 1 it won't be visible that the uncertainty is higher at the long station locations and that the vice versa is true for Exp.5. Nevertheless, we have included below Figure 14 with a same color scale for all durations and experiments.

- **Comment 5:** Typos
  P7 L223 "Wakely" should be "Wakeby"
  EQ 5 fix the parenthesis
  P14 L428 "In contract" should be "in contrast"

  **Response**: Thank you for pointing out the typos. We have fixed them.

With kind regards,

Bora Shehu

[Figure]

***Figure 14*** *The precision (nCI95[%]) in estimation rainfall depth at different durations and 100 year return period for whole Germany with all available data for three experiments: upper row – results obtained from the propagation of 100 local resampled data to the final regionalisation (Exp. 1), middle row –results obtained from 100 spatial simulations of KED[LS|SS] (Exp. 4), lower row – results obtained from 10,000 local resampling and spatial simulations of KED[LS|SS] (Exp. 5). The black squares indicate the locations of LS, while the black lines illustrate the boundaries of German Federal states. Note that the ranges for the legend colours are changing for each experiment in order to emphasize the spatial structure of each experiment.*

---

## Author Response (AR1)

Dear Editor and Reviewers,

We would like to thank you once again for your work on this manuscript and as well the reviewers for their valuable feedback. Following their comments, the following changes have been implemented in the updated version of the manuscript:

- Lines 5-6, since I have changed my working place, I have updated the affiliation and email correspondence.
- Lines 59-61, following the reviewer 1 comments we have updated the sentence as following: "In this paper, the focus is on developing a framework that accounts for uncertainties due to short observation lengths and non-representativeness of point measurements for spatial dependencies of extremes."
- Lines 90-94, following the reviewer 1 comment 2 we have updated the sentence as following: "It is widely accepted that the kriging system can capture only the local uncertainty (providing information at one location at a time conditioned to other observations in the vicinity) and not the spatial one (providing a measure of uncertainty about the un- sampled values taken altogether in space rather than one by one), the estimated uncertainty is dependable on the data configuration rather than on the value itself, and lastly it fails to preserve the high spatial variability of the target variable"
- Lines 103-105 were added following the comment 2 from reviewer 1:
  "Another advantage of stochastics simulations is the ability to compute directly the confidence intervals for the target variable, while in kriging interpolation the confidence intervals are computed from the kriging variance assuming a normal distribution of the errors."
- Lines 190-197 were added following the comment 1 from reviewer 1:
  "The decision to fix the shape parameter at 0.1 was made based on existing literature and previous analysis that we have conducted on the data set in Germany. For more information regarding the choice of generalisation or shape parameter, the reader is directed to our previous study (Shehu et al., 2022). Keeping the shape parameter as fixed can be a reasonable choice to reduce the high uncertainty that is associated with the extreme values analysis at single stations. As shown in (Shehu et al., 2022), for return period higher than 20 years, the uncertainty from a free shape parameter is much higher than the uncertainty from keeping the shape parameter fixed at 0.1, which will cause the interpolation of extreme rainfall to be less certain."
- Lines 309-311 were added following the comment from reviewer 1:
  "Only 133 values from all the stations were sampled here, to address the uncertainty in computing the variogram from a small dataset that corresponds with the number of the long recording stations that were used to compute the variogram for the KED interpolation."
- Lines 321-325 were added following the comment from reviewer 1:
  "In other words, for the known locations where there are observations, either the nodes are not resampled (as in the case of simulated annealing) or the nodes are allowed to vary according to the variogram nugget when compared to the observations (as in the case of the sequential Gaussian simulation). The spatial simulations are conditioned to the location of the 133 long recording stations (LS) since they are the main input for the regionalization are considered the ground truth."
- Lines 377-378 were updated following the comment 3 from reviewer 2:

"Redo step 3-4, until a maximum number of swaps is reached, or if a maximum number of accepted swaps is reached. If this is the case, the temperature t is reduced by a multiplicative factor Ω (here as 0.1)."

- Lines 474-475 were added following the comment 3 from reviewer 1:
  "The parameters are varying greatly in space, and that is why when sampling from space (spatial simulations) the prediction intervals are higher than for the bootstrapping case (or the other cases)."

- Lines 521-526 were added following the comment 2 from reviewer 2:
  "Some outliers are present in the accuracy plot (lower row Figure 10) however expect for one location, these outliers are within the maximum RMSE manifested by the direct regionalisation. The behaviour of these outliers emerges both from parameter outliers and from looking at the quantiles. They are present at locations where parameters are considerably different from the neighbour long observations (as in the case of singular stations in the Black Forest or the Alps), or where a parameter outlier is located (as in the case of Münster City where a very rare extreme event in 2014 causes a high value for the scale σ parameter) and are not geographically clustered."

- Line 534 was updated following the comment from reviewer 1:
  "This parabolic behaviour over the different duration levels is…"

- Figure 12 was updated following the comments of reviewer 1 (see below).

- We have corrected all the technical comments suggested by both reviewer 1 and 2.

- We have not change Figure 14. Actually, the reviewer 2 has advised to have the same colour scaling for the three experiments and durations, however we find that the spatial structure, which is important to see, will be lost. In our response to the reviewer #2 we have inserted a version of the figure with same colour scaling; however we would prefer to keep the Figure 14 as it is in the actual version of the manuscript.

- Throughout the manuscript we have tried to keep the term consistent for LS as long recording stations and for SS as short recording stations.

with kind regards,

Bora Shehu

[Figure]

*Figure 12 Examples of DDF estimates from observed data and predicted by simulations of Exp. 4 and 5 in a cross-validation mode: as median over all simulations and as 95% tolerance ranges from all simulations: upper row for return period T=1years, middle row for T=10years and lower row for return period T=100years. Three stations are shown here: K000830 located in the German-Alps, KO00490 location in Lower Saxony, and KO00550 located in Black Forest.*